# Embedding Principle of Loss Landscape of Deep Neural Networks

**Yaoyu Zhang**[1,2][*] **Zhongwang Zhang**[1] [†] **Tao Luo**[1], **Zhi-Qin John Xu**[1][‡]
[1] School of Mathematical Sciences, Institute of Natural Sciences, MOE-LSC and
Qing Yuan Research Institute, Shanghai Jiao Tong University
[2] Shanghai Center for Brain Science and Brain-Inspired Technology
{zhyy.sjtu, 0123zzw666, luotao41, xuzhiqin}@sjtu.edu.cn.

## Abstract

Understanding the structure of loss landscape of deep neural networks (DNNs) is obviously important. In this work, we prove an embedding principle that the loss landscape of a DNN "contains" all the critical points of all the narrower DNNs. More precisely, we propose a critical embedding such that any critical point, e.g., local or global minima, of a narrower DNN can be embedded to a critical point/affine subspace of the target DNN with higher degeneracy and preserving the DNN output function. Note that, given any training data, differentiable loss function and differentiable activation function, this embedding structure of critical points holds. This general structure of DNNs is starkly different from other nonconvex problems such as protein-folding. Empirically, we find that a wide DNN is often attracted by highly-degenerate critical points that are embedded from narrow DNNs. The embedding principle provides a new perspective to study the general easy optimization of wide DNNs and unravels a potential implicit low-complexity regularization during the training. Overall, our work provides a skeleton for the study of loss landscape of DNNs and its implication, by which a more exact and comprehensive understanding can be anticipated in the near future.

## 1 Introduction

Understanding the loss landscape of DNNs is essential for a theory of deep learning. An important problem is to quantify exactly how the loss landscape looks like (E et al., 2020). This problem is difficult since the loss landscape is so complicated that it can almost be any pattern (Skorokhodov and Burtsev, 2019). Moreover, its high dimensionality and the dependence on data, model and loss make it very difficult to obtain a general understanding through empirical study. Therefore, though it has been extensively studied over the years, it remains an open problem to provide a clear picture about the organization of its critical points and their properties.

In this work, we make a step towards this goal through proposing a very general embedding operation of network parameters from narrow to wide DNNs, by which we prove an embedding principle for fully-connected DNNs stated *intuitively* as follows:

***Embedding principle***: *the loss landscape of any network "contains" all critical points of all narrower networks.*

---

[*]Corresponding author: zhyy.sjtu@sjtu.edu.cn.

[†]Part of this work is done when ZZ was an undergraduate student of Zhiyuan Honors Program at Shanghai Jiao Tong University.

[‡]Corresponding author: xuzhiqin@sjtu.edu.cn.

35th Conference on Neural Information Processing Systems (NeurIPS 2021).

A "narrower network" means a DNN of the same depth but width of each layer no larger than the target DNN. The embedding principle slightly abuses the notion of "contain" since parameter space of DNNs of different widths are different. However, this inclusion relation is reasonable in the sense that, by our embedding operation, any critical point of any narrower network can be embedded to a critical point of the target network preserving its output function. Because of this criticality preserving property, we call this embedding operation the critical embedding.

We conclude our study by a "principle" since the embedding principle is a very general property of loss landscape of DNNs independent of the training data and choice of loss function, and is intrinsic to the layer-wise architecture of DNNs. In addition, the embedding principle is closely related to the training of DNNs. For example, as shown in Fig. 1(a), the training of a width-500 two-layer tanh NN experiences stagnation around the blue dot presumably very close to a saddle point, where the loss decreases extremely slowly. As shown in Fig. 1(b), we find that the DNN output at this blue point (red solid) is very close to the output of the global minimum (black dashed) of the width-1 NN, indicating that the underlying two critical points of two DNNs with different widths have the same output function conforming with the embedding principle. Importantly, this example shows that the training of a wide DNN can indeed experience those critical points from a narrow DNN unraveled by the embedding principle. Moreover, it demonstrates the potential of a transition from a local/global minimum of a narrow NN to a saddle point of a wide NN, which may be the reason underlying the easy optimization of wide NNs.

The embedding principle suggests an underlying mechanism to understand why heavily overparameterized DNNs often generalize well (Breiman, 1995; Zhang et al., 2017) as follows. Roughly, the overparameterized DNN has a large capacity, which seems contradictory to the conventional learning theory, i.e., learning by a model of large capacity easily leads to overfitting. The embedding principle shows that the optima of a wide network intrinsically may be embedded from an optima of a much narrower network, thus, its effective capacity is much smaller. For example, as illustrated in Fig. 1, training of a heavily overparametrized width-500 NN (vs. 50 training data) with small initialization first stagnated around a saddle presumably from width-1 NN and later converges to a global minimum presumably from width-3 NN, which clearly does not overfit. This implicit regularization effect unraveled by the embedding principle is consistent with previous works, such as low-complexity bias (Arpit et al., 2017; Kalimeris et al., 2019; Jin et al., 2020), low-frequency bias (Xu et al., 2019, 2020; Rahaman et al., 2019), and condensation phenomenon of network weights (Luo et al., 2021; Chizat and Bach, 2018; Ma et al., 2020).

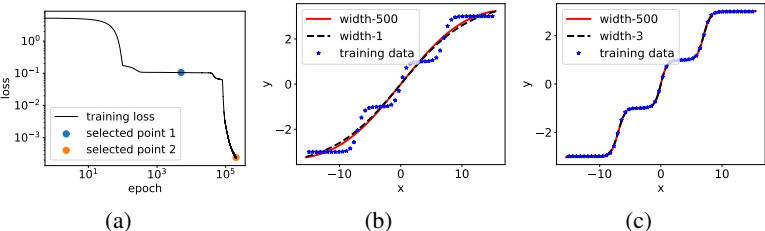

Figure 1: (a) The training loss of two-layer tanh neural network with 500 hidden neurons. (b) (c) Red solid: the DNN output at a training step indicated by (b) the blue dot or (c) the orange dot in (a); Black dashed: the output of the global minimum of (b) width-1 DNN or (c) width-3 DNN, respectively; Blue dots: training data.

## 2  Related works

The loss landscape of DNNs is complex and related to the generalization. Skorokhodov and Burtsev (2019) numerically show that the loss landscape can almost be any pattern. Keskar et al. (2017) visualize minimizers in a 1d slice and suggest that a flat minimizer generalizes better. Wu et al. (2017) find that the volume of basin of attraction of good minima may dominate over that of poor minima in practical problems. He et al. (2019) show that at a local minimum there exist many asymmetric directions such that the loss increases abruptly along one side, and slowly along the opposite side.

Degeneracy is also an important property of minima. Cooper (2021) shows that global minima is typically a high dimensional manifold for overparameterized DNNs. Sagun et al. (2016) empirically shows that Hessian of the minimizer obtained by the training has many zero eigenvalues. Under strong assumptions, Choromanska et al. (2015) shows minima tend to be highly degenerate. This work demonstrates wide existence of highly degenerate critical points, including local or global minima and saddle points, in the loss landscape by the embedding principle.

Lots of previous theoretical works focus on very wide DNNs, such as the phase diagram of two-layer ReLU infinite-width NNs (Luo et al., 2021), NTK regime (Jacot et al., 2018; Arora et al., 2019; Zhang et al., 2020; Du et al., 2019; Zou et al., 2018; Allen-Zhu et al., 2019; E et al., 2019), mean-field regime (Mei et al., 2018; Rotskoff and Vanden-Eijnden, 2018; Chizat and Bach, 2018; Sirignano and Spiliopoulos, 2020). By the embedding principle, this work demonstrate the loss landscape similarity between a moderate-width NN and a very wide NN, that they share a set of critical points embedded from that of narrower NNs. Therefore, results about infinite-width NNs could provide valuable insights about training of finite-width NNs used in practice.

The complexity of NN output increases during the training (Arpit et al., 2017; Xu et al., 2019, 2020; Rahaman et al., 2019; Kalimeris et al., 2019; Goldt et al., 2020; He et al., 2020; Mingard et al., 2019; Jin et al., 2020). For example, the frequency principle (Xu et al., 2019, 2020) states that DNNs often fit target functions from low to high frequencies during the training.

In Zhang et al. (2021), we make a comprehensive extension of this conference paper. In the long paper, we provide a mathematical definition of the critical embedding and propose a new class of general compatible embeddings, which is a much wider class of critical embeddings than composition embeddings in this work. These general compatible embeddings provide much richer details about the geometry of critical submanifolds of DNN loss landscape. Note that the composition embedding technique is also studied in Fukumizu et al. (2019) and Simsek et al. (2021).

## 3 Main results

In this section, we intuitively summarize our key theoretical results about the embedding principle and empirically demonstrate its relevance to practice, starting from proposing an embedding operation as follows. Rigorous theoretical description and proofs are presented in the latter sections.

### 3.1 Characteristics of embedding principle

Consider a neural network $f_{\boldsymbol{\theta}}(\boldsymbol{x})$, where $\boldsymbol{\theta}$ is the set of all network parameters, $\boldsymbol{x} \in \mathbb{R}^d$ is the input. We summarize assumptions and provide definitions needed for all our results in this work below.

**Assumption.** *(i) L-layer ($L \geq 2$) fully-connected NN.*

*(ii) Training data $S = \{(\boldsymbol{x}_i, \boldsymbol{y}_i)\}_{i=1}^n$, $n \in \mathbb{Z}^+ \cup \{+\infty\}$.*

*(iii)Loss function $R_S(\boldsymbol{\theta}) = \mathbb{E}_S \ell(f_{\boldsymbol{\theta}}(\boldsymbol{x}), \boldsymbol{y})$.*

*(iv) Loss function and activation function are differentiable. Note that, even for functions like ReLU or hinge loss, as long as we uniquely assign a subgradient to their non-differentiable points, all our results still hold.*

**Definition 1** (**critical point**). *Parameter vector $\boldsymbol{\theta}$ is a critical point of the landscape of $R_S$ if $\nabla_{\boldsymbol{\theta}} R_S(\boldsymbol{\theta}) = \boldsymbol{0}$.*

**Definition 2** (**critical submanifold/affine subspace**). *A critical submanifold or affine subspace $\mathcal{M}$ is a connected subsubmanifold or affine subspace of the parameter space $\mathbb{R}^M$, such that each $\boldsymbol{\theta} \in \mathcal{M}$ is a critical point of loss with the same loss value.*

**Definition 3** (**degree of degeneracy**). *The degree of degeneracy of point $\boldsymbol{\theta}$ in the landscape of $R_S$ is the corank of Hessian matrix $\nabla_{\boldsymbol{\theta}} \nabla_{\boldsymbol{\theta}} R_S$, i.e., number of the zero eigenvalues.*

**Remark.** *In the above definition of degree of degeneracy, we require twice differentiable activation function and twice differentiable loss to compute Hessian for convenience. For loss and activation functions with only first-order differentiability, we extend the definition of degree of degeneracy as follows: for any critical point $\boldsymbol{\theta}$ belonging to a $K$-dimensional critical submanifold $\mathcal{M}$, its degree of degeneracy is at least $K$.*

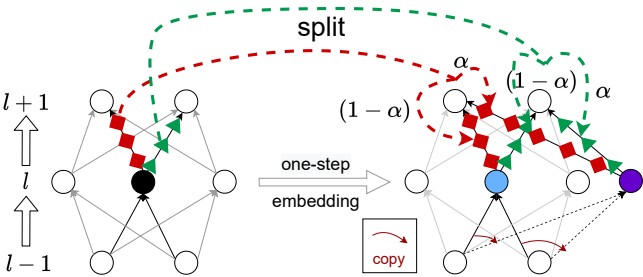

Figure 2: Illustration of one-step embedding. The black neuron in the left network is splitted into the blue and purple neurons in the right network. The red (green) output weight of the black neuron in the left net is splitted into two red (green) weights in the right net with ratio $\alpha$ and $(1 - \alpha)$, respectively.

We first introduce one-step embedding intuitively, and leave the rigorous definition latter. As shown in Fig. 2, an one-step embedding is performed by splitting any hidden neuron, say the black neuron in the left network, into two neurons colored in blue and purple in the right network. The input weights of the two splitted neurons are the same as the input weights of the original black neuron. Each output weight of the original black neuron is splitted into two parts of fraction $\alpha$ and $(1 - \alpha)$ ($\alpha \in \mathbb{R}$, a hyperparameter), respectively. The multi-step embedding is the composition of multiple one-step embeddings. Since each one-step embedding can add one neuron to a selected layer, parameter of any NN can be embedded to the parameter space of any wider NN through a multi-step embedding. The multi-step embedding operation leads to the following property readily.

**Proposition** (**one-step embedding preserves network properties, informal Prop. 1**). *For any point $\boldsymbol{\theta}_{\mathrm{narr}}$ of a DNN, a point $\boldsymbol{\theta}_{\mathrm{wide}}$ of a wider DNN obtained from $\boldsymbol{\theta}_{\mathrm{narr}}$ by one-step embedding satisfies*
*(i) $\boldsymbol{f}_{\boldsymbol{\theta}_{\mathrm{narr}}}(\boldsymbol{x}) = \boldsymbol{f}_{\boldsymbol{\theta}_{\mathrm{wide}}}(\boldsymbol{x})$ for any $\boldsymbol{x}$;*
*(ii) representation of the wide DNN at $\boldsymbol{\theta}_{\mathrm{wide}}$, i.e., the set of all different response functions of neurons, is the same as representation of the narrow DNN at $\boldsymbol{\theta}_{\mathrm{narr}}$.*

The most important property of this embedding is criticality preserving as follows.

**Theorem** (**criticality preserving, informal Theorem 1**). *For any critical point $\boldsymbol{\theta}_{\mathrm{narr}}$ of a DNN, a point $\boldsymbol{\theta}_{\mathrm{wide}}$ of a wider DNN obtained from $\boldsymbol{\theta}_{\mathrm{narr}}$ by multi-step embedding is a critical point.*

The embedding operation explains the cause of a type of degeneracy in the loss landscape.

**Theorem** (**degeneracy of embedded critical points, informal Theorem 2**). *If output weights of neurons in each layer of a DNN at a critical point $\boldsymbol{\theta}_{\mathrm{narr}}$ are not all zero, then, for any $K$-neuron wider DNN, $\boldsymbol{\theta}_{\mathrm{narr}}$ can be embedded to a $K$-dimensional critical affine subspace.*

**Remark.** *By above theorem, each step of embedding of a critical point in general is accompanied by an increased degree of degeneracy. Therefore, degenerate critical points in general widely exist in the loss landscape of a DNN, and non-degenerate critical points are rare because they often become degenerate once embedded to a wider DNN.*

In previous studies, degeneracy is often considered as a consequence of over-parameterization depending on the size of training data $n$. Specifically, Cooper (2021) proves that the degree of degeneracy of global minima is $m - n$ for 1-d output, where $m$ is the number of network parameters. However, we demonstrate by the above theorem that regardless of whether the NN is over-parameterized, degenerate critical points are prevalent in its loss landscape as long as narrower DNNs possess critical points.

### 3.2 Numerical experiments

**Experimental setup.** Throughout this work, we use two-layer fully-connected neural network with size $d$-$m$-$d_{out}$. The input dimension $d$ is determined by the training data. The output dimension $d_{out}$ is different for different experiments. The number of hidden neurons $m$ is specified in each experiment. All parameters are initialized by a Gaussian distribution with mean zero and variance specified in each experiment. We use MSE loss trained by full batch gradient descent for 1D fitting

problems (Figs. 1, 3(a) and 4), and default Adam optimizer with full batch for others. The learning rate is fixed throughout the training. More details of experiments are shown in Appendix B.

**Increment of degeneracy through embedding.** We train a two-layer NN of width $m_{\text{small}} = 2$ to learn data of Fig. 1 shown in Fig. 3(a) or Iris dataset (Fisher, 1936) in Fig. 3(b) to a critical point. We first roughly estimate the possible interval of critical points by observing where the loss decays very slowly, and then take the point with the smallest derivative of the parameters (use $L_1$ norm) as an empirical critical point. The $L_1$ norm of the derivative of loss function at the empirical critical point is approximately $7.15 \times 10^{-15}$ for Fig. 3(a) and $3.72 \times 10^{-13}$ for Fig. 3(b), which are reasonably small. We then embed this critical point to networks of width $m = 3$ and $m = 4$ through an one-step or a two-step embedding, respectively. It is obvious from Fig. 3 that each step of embedding incurs one more zero eigenvalue in the Hessian matrix, which conforms with Theorem 2. Moreover, in Fig. 3(a), for $m = 2$, all eigenvalues are positive (red) indicating the critical point obtained by training is a local or global minimum. After embedding, this point becomes a saddle due to the emergence of negative eigenvalues (blue). Specifically, in both Fig. 3(a) and (b), we observe a steady increase of significant negative eigenvalues, e.g., from 0 to 1 to 2 in (a) and from 3 to 5 to 7 in (b), which implies reduced difficulty in escaping from the corresponding critical point in a wider NN during the training.

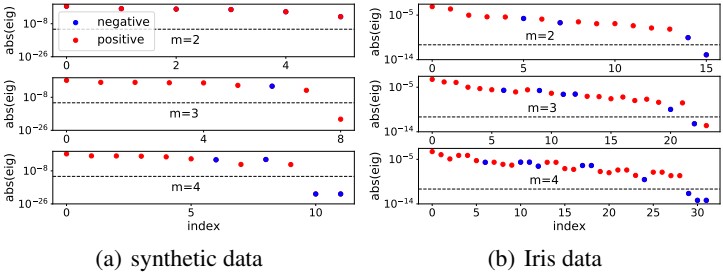

(a) synthetic data          (b) Iris data

Figure 3: Eigenvalues of Hessian of NNs at the critical points embedded from the NN with width $m_{\text{small}} = 2$ for learning data of Fig. 1 in (a) and for Iris dataset in (b). The value of $m$ in each sub-figure is the NN width after embedding. The auxiliary dash line in each sub-figure is $y = 10^{-11}$. We equally split one neuron of a width-2 two-layer NN at a critical point into k neurons ($k = 2, 3$), whose input weights remain the same but output weights are $1/k$ of the original neuron.

**Empirical diagram of loss landscape.** In Fig. 4, we present an empirical diagram of loss landscape of a width-3 two-layer tanh DNN to visualize a set of its critical points predicted by the embedding principle, i.e., critical points embedded from network of width-1 or -2 respectively as well as critical points that cannot be obtained through embedding. Through the training of width-1, -2, -3 network respectively on the training data presented in Fig. 1 for multiple trials, we discover 1 critical point for width-1 network, 2 critical points for width-2 network and 1 critical point for width-3 network that cannot be embedded from a narrower NN. Then, embedding all these four critical points to critical points/affine subspaces of loss landscape of the width-3 network, we obtain four sets of critical points with their loss values, output functions, degrees of degeneracy and width of network they embedded from illustrated in Fig. 4. This diagram immediately tells us what attracts the gradient-based training trajectory for a width-3 network. Specifically, if stagnation happens during the training, this diagram informs us the potential loss values and output functions at stagnation, which could help us better understand the nonlinear training process of not only a width-3 network but also much wider networks due to the embedding principle. Furthermore, as illustrated in Fig. 1 for the training process of a 500-neuron NN, saddle points of a wide NN, effectively local or global minima of narrow NNs, composes a trajectory, which may serve as a compass for achieving a global minimum from narrow NNs of low complexity.

**Reduction of DNN at critical point.** The embedding principle predicts a class of critical points of a NN embedded from much narrower NNs. At such a critical point, we shall be able to find neuron groups, within which neurons have similar orientation of input weights presumably originated from the same neuron of a narrow NN through embedding. This prediction is confirmed by the following experiment in Fig. 5. We train a width-400 two-layer ReLU NN $\boldsymbol{f_\theta} = \sum_{k=1}^{m} a_k \sigma(\boldsymbol{w}_k^T \tilde{\boldsymbol{x}})$

**empirical diagram of loss landscape**

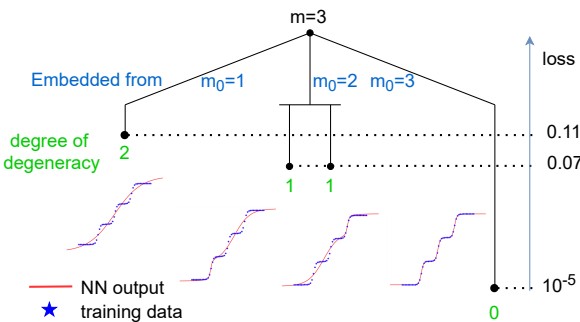

Figure 4: Empirical diagram of loss landscape of a width-3 two-layer tanh NN, i.e., all critical points width-3 or narrower NNs may get close to during the training under proper initialization. Each black dot at terminal represents a specific set of critical points of loss embedded from critical points of NNs of different widths (blue). These critical points have different loss values (ordinate), degrees of degeneracy (green) and output functions (red solid curves) as labelled in the figure. The blue dots represent the training data. We use the same equal splitting as Fig. 3 to embed critical points of width-1 or width-2 NN to critical points of the width-3 NN and compute the hessian to obtain the corresponding degree of degeneracy. Note that the degree of degeneracy of these critical points computed numerically in this problem coincides with their minimal degree of degeneracy $m - m_0$ in Theorem 2.

($\tilde{\boldsymbol{x}} = [\boldsymbol{x}^\mathsf{T}, 1]^\mathsf{T}$) on 1000 training samples of the MNIST dataset with small initialization. At the blue dot in Fig. 5(a), the loss decreases very slowly, presumably very close to a saddle point. We then examine the orientation similarity between each pair of neuron input weights by computing the inner product of two normalized input weight. As shown in Fig. 5(b), there emerge 58 groups of neurons (neurons with very small amplitudes are neglected and later directly removed), where similarity between input weights in the same group is at least 0.9. For each group $S_{\text{similar}}$, we randomly select a neuron $j$, replace its output weight by $\sum_{k \in S_{\text{similar}}} a_k \|\boldsymbol{w}_k\|_2 / \|\boldsymbol{w}_j\|_2$, and discard all other neurons in the group. The parameter set before reduction is denoted by $\boldsymbol{\theta}_{\text{ori}}$, and after reduction by $\boldsymbol{\theta}_{\text{redu}}$. Width of the NN is reduced from 400 to 58. We train the reduced NN from $\boldsymbol{\theta}_{\text{redu}}$ as shown in Fig. 5, which stagnated after a few steps at the same loss value as the blue point in Fig. 5(a) marked by the blue dash and represented by the blue point in Fig. 5(c). We then compare the prediction between original model and the reduced model at the corresponding blue points on 10000 test data as shown in Fig.5(d). For each grid, color indicates the frequency of that prediction pair. Specifically, the highlight of diagonal element indicts high prediction agreement of two models (overall $\sim 98.5\%$). Therefore, this critical point of the reduced width-58 NN well matches the critical point of the original width-400 NN, clearly demonstrating the relevance of our embedding principle to real dataset training.

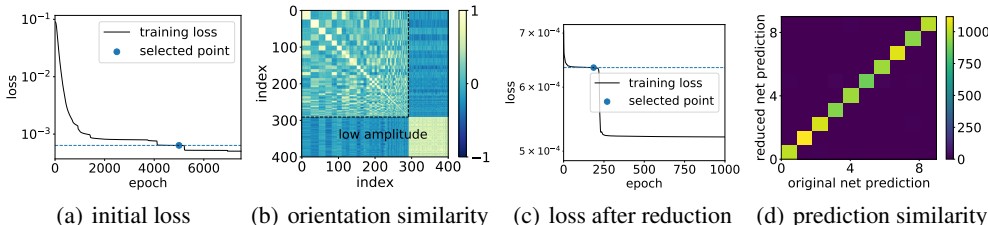

(a) initial loss    (b) orientation similarity    (c) loss after reduction    (d) prediction similarity

Figure 5: (a) The training loss of the initial network on MNIST. The blue point is selected for reduction. (b) The normalized inner product of input weights for different neurons. The abscissa and ordinate represent neuron index. Neurons in "low amplitude" region has much lower amplitude than others, hence are removed. (c) The training loss of the reduced network. Blue dash indicates the same loss value as the blue dash in (a). The blue point is selected as a representative for comparison. (d) Prediction similarity. For each grid, color indicates the frequency of that prediction pair.

# 4 Preliminaries

## 4.1 Deep Neural Networks

Consider $L$-layer ($L \geq 2$) fully-connected DNNs with a general differentiable activation function. We regard the input as the 0-th layer and the output as the $L$-th layer. Let $m_l$ be the number of neurons in the $l$-th layer. In particular, $m_0 = d$ and $m_L = d'$. For any $i, k \in \mathbb{N}$ and $i < k$, we denote $[i:k] = \{i, i+1, \ldots, k\}$. In particular, we denote $[k] := \{1, 2, \ldots, k\}$. Given weights $W^{[l]} \in \mathbb{R}^{m_l \times m_{l-1}}$ and bias $b^{[l]} \in \mathbb{R}^{m_l}$ for $l \in [L]$, we define the collection of parameters $\boldsymbol{\theta}$ as a $2L$-tuple (an ordered list of $2L$ elements) whose elements are matrices or vectors

$$\boldsymbol{\theta} = \left(\boldsymbol{\theta}|_1, \cdots, \boldsymbol{\theta}|_L\right) = \left(\boldsymbol{W}^{[1]}, \boldsymbol{b}^{[1]}, \ldots, \boldsymbol{W}^{[L]}, \boldsymbol{b}^{[L]}\right). \tag{1}$$

where the $l$-th layer parameters of $\boldsymbol{\theta}$ is the ordered pair $\boldsymbol{\theta}|_l = \left(\boldsymbol{W}^{[l]}, \boldsymbol{b}^{[l]}\right)$, $l \in [L]$. We may misuse of notation and identify $\boldsymbol{\theta}$ with its vectorization $\text{vec}(\boldsymbol{\theta}) \in \mathbb{R}^M$ with $M = \sum_{l=0}^{L-1}(m_l + 1)m_{l+1}$.

Given $\boldsymbol{\theta} \in \mathbb{R}^M$, the neural network function $\boldsymbol{f_\theta}(\cdot)$ is defined recursively. First, we write $\boldsymbol{f}_{\boldsymbol{\theta}}^{[0]}(\boldsymbol{x}) = \boldsymbol{x}$ for all $\boldsymbol{x} \in \mathbb{R}^d$. Then for $l \in [L-1]$, $\boldsymbol{f}_{\boldsymbol{\theta}}^{[l]}$ is defined recursively as $\boldsymbol{f}_{\boldsymbol{\theta}}^{[l]}(\boldsymbol{x}) = \sigma(\boldsymbol{W}^{[l]}\boldsymbol{f}_{\boldsymbol{\theta}}^{[l-1]}(\boldsymbol{x}) + \boldsymbol{b}^{[l]})$. Finally, we denote

$$\boldsymbol{f_\theta}(\boldsymbol{x}) = \boldsymbol{f}(\boldsymbol{x}, \boldsymbol{\theta}) = \boldsymbol{f}_{\boldsymbol{\theta}}^{[L]}(\boldsymbol{x}) = \boldsymbol{W}^{[L]}\boldsymbol{f}_{\boldsymbol{\theta}}^{[L-1]}(\boldsymbol{x}) + \boldsymbol{b}^{[L]}. \tag{2}$$

For notational simplicity, we may drop the subscript $\boldsymbol{\theta}$ in $\boldsymbol{f}_{\boldsymbol{\theta}}^{[l]}$, $l \in [0:L]$.

We introduce the following notions for the convenience of the presentation in this paper.

**Definition 4 (Wider/narrower DNN).** *We write* $\text{NN}(\{m_l\}_{l=0}^L)$ *for a fully-connected neural network with width* $(m_0, \ldots, m_L)$. *Given two $L$-layer ($L \geq 2$) fully-connected neural networks* $\text{NN}(\{m_l\}_{l=0}^L)$ *and* $\text{NN}'(\{m_l'\}_{l=0}^L)$, *if* $m_0' = m_0$, $m_L' = m_L$, *and for any* $l \in [L-1]$, $m_l' \geq m_l$ *and* $K = \sum_{l=1}^{L-1}(m_l' - m_l) \in \mathbb{N}_+$, *then we say that* $\text{NN}'(\{m_l'\}_{l=0}^L)$ *is $K$-neuron wider than* $\text{NN}(\{m_l\}_{l=0}^L)$ *and* $\text{NN}(\{m_l\}_{l=0}^L)$ *$K$-neuron narrower than* $\text{NN}'(\{m_l'\}_{l=0}^L)$.

## 4.2 Loss function

The training data set is denoted as $S = \{(\boldsymbol{x}_i, \boldsymbol{y}_i)\}_{i=1}^n$, where $\boldsymbol{x}_i \in \mathbb{R}^d$, $\boldsymbol{y}_i \in \mathbb{R}^{d'}$. For simplicity, here we assume an unknown function $\boldsymbol{y}$ satisfying $\boldsymbol{y}(\boldsymbol{x}_i) = \boldsymbol{y}_i$ for $i \in [n]$. The empirical risk reads as

$$R_S(\boldsymbol{\theta}) = \frac{1}{n}\sum_{i=1}^n \ell(\boldsymbol{f}(\boldsymbol{x}_i, \boldsymbol{\theta}), \boldsymbol{y}(\boldsymbol{x}_i)) = \mathbb{E}_S \ell(\boldsymbol{f}(\boldsymbol{x}, \boldsymbol{\theta}), \boldsymbol{y}). \tag{3}$$

where the expectation $\mathbb{E}_S h(\boldsymbol{x}) := \frac{1}{n}\sum_{i=1}^n h(\boldsymbol{x}_i)$ for any function $h : \mathbb{R}^d \to \mathbb{R}$ and the loss function $\ell(\cdot, \cdot)$ is differentiable and the derivative of $\ell$ with respect to its first argument is denoted by $\nabla \ell(\boldsymbol{y}, \boldsymbol{y}^*)$. Generally, we always take derivatives/gradients of $\ell$ in its first argument with respect to any parameter. We consider gradient flow of $R_S$ as the training dynamics, i.e., $\mathrm{d}\boldsymbol{\theta}/\mathrm{d}t = -\nabla_{\boldsymbol{\theta}} R_S(\boldsymbol{\theta})$ with $\boldsymbol{\theta}(0) = \boldsymbol{\theta}_0$.

We define the error vectors $\boldsymbol{z}_{\boldsymbol{\theta}}^{[l]} = \nabla_{\boldsymbol{f}^{[l]}} \ell$ for $l \in [L]$ and the feature gradients $\boldsymbol{g}_{\boldsymbol{\theta}}^{[L]} = \mathbf{1}$ and $\boldsymbol{g}_{\boldsymbol{\theta}}^{[l]} = \sigma^{(1)}\left(\boldsymbol{W}^{[l]}\boldsymbol{f}_{\boldsymbol{\theta}}^{[l-1]} + \boldsymbol{b}^{[l]}\right)$ for $l \in [L-1]$. Here $\sigma^{(1)}$ is the first derivative of $\sigma$. We call $\boldsymbol{f}_{\boldsymbol{\theta}}^{[l]}$, $l \in [L]$ feature vectors. The collections of feature vectors, feature gradients, and error vectors are $\boldsymbol{F_\theta} = \{\boldsymbol{f}_{\boldsymbol{\theta}}^{[l]}\}_{l=1}^L$, $\boldsymbol{G_\theta} = \{\boldsymbol{g}_{\boldsymbol{\theta}}^{[l]}\}_{l=1}^L$, $\boldsymbol{Z_\theta} = \{\boldsymbol{z}_{\boldsymbol{\theta}}^{[l]}\}_{l=1}^L$. Using backpropagation, we can calculate the gradients as follows

$$\boldsymbol{z}_{\boldsymbol{\theta}}^{[L]} = \nabla \ell, \quad \boldsymbol{z}_{\boldsymbol{\theta}}^{[l]} = (\boldsymbol{W}^{[l+1]})^\mathsf{T} \boldsymbol{z}_{\boldsymbol{\theta}}^{[l+1]} \circ \boldsymbol{g}_{\boldsymbol{\theta}}^{[l+1]}, \quad l \in [L-1],$$
$$\nabla_{\boldsymbol{W}^{[l]}} \ell = \boldsymbol{z}_{\boldsymbol{\theta}}^{[l]} \circ \boldsymbol{g}_{\boldsymbol{\theta}}^{[l]}(\boldsymbol{f}_{\boldsymbol{\theta}}^{[l-1]})^\mathsf{T}, \quad \nabla_{\boldsymbol{b}^{[l]}} \ell = \boldsymbol{z}_{\boldsymbol{\theta}}^{[l]} \circ \boldsymbol{g}_{\boldsymbol{\theta}}^{[l]}, \quad l \in [L].$$

Here we use $\circ$ for the Hadamard product of two matrices of the same dimension.

# 5 Critical embedding

We introduce the one-step embedding for the DNNs which will lead us to general embeddings.

**Definition 5** (**one-step embedding**). *Given a L-layer ($L \geq 2$) fully-connected neural network with width $(m_0, \ldots, m_L)$ and network parameters $\boldsymbol{\theta} = (\boldsymbol{W}^{[1]}, \boldsymbol{b}^{[1]}, \cdots, \boldsymbol{W}^{[L]}, \boldsymbol{b}^{[L]}) \in \mathbb{R}^M$, for any $l \in [L-1]$ and any $s \in [m_l]$, we define the linear operators $\mathcal{T}_{l,s}$ and $\mathcal{V}_{l,s}$ applying on $\boldsymbol{\theta}$ as follows*

$$\mathcal{T}_{l,s}(\boldsymbol{\theta})|_k = \boldsymbol{\theta}|_k, \quad k \neq l, l+1,$$

$$\mathcal{T}_{l,s}(\boldsymbol{\theta})|_l = \left( \begin{bmatrix} \boldsymbol{W}^{[l]} \\ \boldsymbol{W}^{[l]}_{s,[1:m_{l-1}]} \end{bmatrix}, \begin{bmatrix} \boldsymbol{b}^{[l]} \\ \boldsymbol{b}^{[l]}_s \end{bmatrix} \right), \quad \mathcal{T}_{l,s}(\boldsymbol{\theta})|_{l+1} = \left( \begin{bmatrix} \boldsymbol{W}^{[l+1]}, \boldsymbol{0}_{m_{l+1} \times 1} \end{bmatrix}, \boldsymbol{b}^{[l+1]} \right),$$

$$\mathcal{V}_{l,s}(\boldsymbol{\theta})|_k = \left( \boldsymbol{0}_{m_k \times m_{k-1}}, \boldsymbol{0}_{m_k \times 1} \right), \quad k \neq l, l+1,$$

$$\mathcal{V}_{l,s}(\boldsymbol{\theta})|_l = \left( \boldsymbol{0}_{(m_l+1) \times m_{l-1}}, \boldsymbol{0}_{(m_l+1) \times 1} \right),$$

$$\mathcal{V}_{l,s}(\boldsymbol{\theta})|_{l+1} = \left( \begin{bmatrix} \boldsymbol{0}_{m_{l+1} \times (s-1)}, -\boldsymbol{W}^{[l+1]}_{[1:m_{l+1}],s}, \boldsymbol{0}_{m_{l+1} \times (m_l-s)}, \boldsymbol{W}^{[l+1]}_{[1:m_{l+1}],s} \end{bmatrix}, \boldsymbol{0}_{m_{l+1} \times 1} \right).$$

*Then the one-step embedding operator $\mathcal{T}^{\alpha}_{l,s}$ is defined as for any $\boldsymbol{\theta} \in \mathbb{R}^M$*

$$\mathcal{T}^{\alpha}_{l,s}(\boldsymbol{\theta}) = (\mathcal{T}_{l,s} + \alpha \mathcal{V}_{l,s})(\boldsymbol{\theta}).$$

*Note that the resulting parameter $\mathcal{T}^{\alpha}_{l,s}(\boldsymbol{\theta})$ corresponds to a L-layer fully-connected neural network with width $(m_0, \ldots, m_{l-1}, m_l + 1, m_{l+1}, \ldots, m_L)$.*

An illustration of $\mathcal{T}_{l,s}$, $\mathcal{V}_{l,s}$, and $\mathcal{T}^{\alpha}_{l,s}$ can be found in Fig. S1 in Appendix.

**Lemma 1.** *Given a L-layer ($L \geq 2$) fully-connected neural network with width $(m_0, \ldots, m_L)$, for any network parameters $\boldsymbol{\theta} = (\boldsymbol{W}^{[1]}, \boldsymbol{b}^{[1]}, \cdots, \boldsymbol{W}^{[L]}, \boldsymbol{b}^{[L]})$ and for any $l \in [L-1]$, $s \in [m_l]$, we have the expressions for $\boldsymbol{\theta}' := \mathcal{T}^{\alpha}_{l,s}(\boldsymbol{\theta})$*

*(i) feature vectors in $\boldsymbol{F}_{\boldsymbol{\theta}'}$: $\boldsymbol{f}^{[l']}_{\boldsymbol{\theta}'} = \boldsymbol{f}^{[l']}_{\boldsymbol{\theta}}$, $l' \neq l$ and $\boldsymbol{f}^{[l]}_{\boldsymbol{\theta}'} = \left[ (\boldsymbol{f}^{[l]}_{\boldsymbol{\theta}})^{\mathsf{T}}, (\boldsymbol{f}^{[l]}_{\boldsymbol{\theta}})_s \right]^{\mathsf{T}}$;*

*(ii) feature gradients in $\boldsymbol{G}_{\boldsymbol{\theta}'}$: $\boldsymbol{g}^{[l']}_{\boldsymbol{\theta}'} = \boldsymbol{g}^{[l']}_{\boldsymbol{\theta}}$, $l' \neq l$ and $\boldsymbol{g}^{[l]}_{\boldsymbol{\theta}'} = \left[ (\boldsymbol{g}^{[l]}_{\boldsymbol{\theta}})^{\mathsf{T}}, (\boldsymbol{g}^{[l]}_{\boldsymbol{\theta}})_s \right]^{\mathsf{T}}$;*

*(iii) error vectors in $\boldsymbol{Z}_{\boldsymbol{\theta}'}$: $\boldsymbol{z}^{[l']}_{\boldsymbol{\theta}'} = \boldsymbol{z}^{[l']}_{\boldsymbol{\theta}}$, $l' \neq l$*
*and $\boldsymbol{z}^{[l]}_{\boldsymbol{\theta}'} = \left[ (\boldsymbol{z}^{[l]}_{\boldsymbol{\theta}})^{\mathsf{T}}_{[1:s-1]}, (1-\alpha)(\boldsymbol{z}^{[l]}_{\boldsymbol{\theta}})_s, (\boldsymbol{z}^{[l]}_{\boldsymbol{\theta}})^{\mathsf{T}}_{[s+1:m_l]}, \alpha(\boldsymbol{z}^{[l]}_{\boldsymbol{\theta}})_s \right]^{\mathsf{T}}$.*

An illustration of $\boldsymbol{F}_{\boldsymbol{\theta}}$ and $\boldsymbol{Z}_{\boldsymbol{\theta}}$ can be found in Fig. S2 in Appendix.

**Proposition 1** (**one-step embedding preserves network properties**). *Given a L-layer ($L \geq 2$) fully-connected neural network with width $(m_0, \ldots, m_L)$, for any network parameters $\boldsymbol{\theta} = (\boldsymbol{W}^{[1]}, \boldsymbol{b}^{[1]}, \cdots, \boldsymbol{W}^{[L]}, \boldsymbol{b}^{[L]})$ and for any $l \in [L-1]$, $s \in [m_l]$, the following network properties are preserved for $\boldsymbol{\theta}' = \mathcal{T}^{\alpha}_{l,s}(\boldsymbol{\theta})$:*

*(i) output function is preserved: $f_{\boldsymbol{\theta}'}(\boldsymbol{x}) = f_{\boldsymbol{\theta}}(\boldsymbol{x})$ for all $\boldsymbol{x}$;*

*(ii) empirical risk is preserved: $R_S(\boldsymbol{\theta}') = R_S(\boldsymbol{\theta})$;*

*(iii) the sets of features are preserved: $\left\{ \left( \boldsymbol{f}^{[l]}_{\boldsymbol{\theta}'} \right)_i \right\}_{i \in [m_l+1]} = \left\{ \left( \boldsymbol{f}^{[l]}_{\boldsymbol{\theta}} \right)_i \right\}_{i \in [m_l]}$ and*

$\left\{ \left( \boldsymbol{f}^{[l']}_{\boldsymbol{\theta}'} \right)_i \right\}_{i \in [m_{l'}]} = \left\{ \left( \boldsymbol{f}^{[l']}_{\boldsymbol{\theta}} \right)_i \right\}_{i \in [m_{l'}]}$ *for $l' \in [L]\backslash\{l\}$;*

**Theorem 1** (**criticality preserving**). *Given a L-layer ($L \geq 2$) fully-connected neural network with width $(m_0, \ldots, m_L)$, for any network parameters $\boldsymbol{\theta} = (\boldsymbol{W}^{[1]}, \boldsymbol{b}^{[1]}, \cdots, \boldsymbol{W}^{[L]}, \boldsymbol{b}^{[L]})$ and for any $l \in [L-1]$, $s \in [m_l]$, if $\nabla_{\boldsymbol{\theta}} R_S(\boldsymbol{\theta}) = \boldsymbol{0}$, then $\nabla_{\boldsymbol{\theta}} R_S(\boldsymbol{\theta}') = \boldsymbol{0}$.*

**Lemma 2** (**increment of the degree of degeneracy**). *Given a L-layer ($L \geq 2$) fully-connected neural network with width $(m_0, \ldots, m_L)$, if there exists $l \in [L-1]$, $s \in [m_l]$, and a q-dimensional manifold $\mathcal{M}$ consisting of critical points of $R_S$ such that for any $\boldsymbol{\theta} \in \mathcal{M}$, $\boldsymbol{W}^{[l+1]}_{[1:m_{l+1}],s} \neq \boldsymbol{0}$, then $\mathcal{M}' := \{\mathcal{T}^{\alpha}_{l,s}(\boldsymbol{\theta}) | \boldsymbol{\theta} \in \mathcal{M}, \alpha \in \mathbb{R}\}$ is a $(q+1)$-dimensional manifold consists of critical points for the corresponding L-layer fully-connected neural network with width $(m_0, \ldots, m_{l-1}, m_l + 1, m_{l+1}, \ldots, m_L)$.*

**Theorem 2 (degeneracy of embedded critical points).** *Consider two L-layer ($L \geq 2$) fully-connected neural networks $\mathrm{NN}_A(\{m_l\}_{l=0}^L)$ and $\mathrm{NN}_B(\{m'_l\}_{l=0}^L)$ which is K-neuron wider than $\mathrm{NN}_A$. Suppose that the critical point $\boldsymbol{\theta}_A = (\boldsymbol{W}^{[1]}, \boldsymbol{b}^{[1]}, \cdots, \boldsymbol{W}^{[L]}, \boldsymbol{b}^{[L]})$ satisfies $\boldsymbol{W}^{[l]} \neq \boldsymbol{0}$ for each layer $l \in [L]$. Then the parameters $\boldsymbol{\theta}_A$ of $\mathrm{NN}_A$ can be critically embedded to a K-dimensional critical affine subspace $\mathcal{M}_B = \{\boldsymbol{\theta}_B + \sum_{i=1}^K \alpha_i \boldsymbol{v}_i | \alpha_i \in \mathbb{R}\}$ of loss landscape of $\mathrm{NN}_B$. Here $\boldsymbol{\theta}_B = (\prod_{i=1}^K \mathcal{T}_{l_i, s_i})(\boldsymbol{\theta}_A)$ and $\boldsymbol{v}_i = \mathcal{T}_{l_K, s_K} \cdots \mathcal{V}_{l_i, s_i} \cdots \mathcal{T}_{l_1, s_1} \boldsymbol{\theta}_A$.*

Note that neuron-index permutation among the same layer is a trivial criticality invariant transform. More discussions about it, specifically for NNs of homogeneous activation functions like ReLU, can be found in Section A.1 in Appendix.

## 6  Conclusion and discussion

In this work, we prove an embedding principle that loss landscape of a DNN *contains* all critical points of all the narrower DNNs. This embedding principle unravels wide existence of highly degenerate critical points with low complexity in the loss landscape of a wide DNN, i.e., critical points with low-complexity output function and degenerate Hessian matrix, embedded from critical points of narrow DNNs. With such a loss landscape of DNN, the gradient-based training has the potential of getting attracted or even converging to a low complexity critical point as confirmed by above numerical experiments, which implies a potential implicit regularization towards low-complexity function of nonlinear DNN training dynamics.

Moreover, through critical embedding, a critical point in form of a common non-degenerate local minimum of a narrow DNN not only becomes degenerate in general, but also may become a saddle point as illustrated by numerical experiments. This may be the reason underlying the general easy optimization of wide DNNs observed in practice even beyond the linear/kernel/NTK regime (Chen et al., 2020; Trager et al., 2019; Geiger et al., 2020; Fort et al., 2020; Luo et al., 2021). We will perform more detailed analysis as well as numerical experiments specifically about this minimum-to-saddle transition later.

At the essence, the embedding principle results from the layer structure of a neural network model, which allows arbitrary neuron addition, input weight copying and output weight splitting within each layer. Therefore, though results in this work assume fully-connected NNs, these can be easily extended to other DNN architectures. Considering convolutional neural networks for example, the quantity that corresponds to width of fully-connected NNs is channel. Similar to one-step or multi-step embedding, we can introduce a feature splitting operation, i.e., increase the number of channels by splitting all neurons sharing one convolution kernel with the same $\alpha$, which can be proven to preserve the output function, representation as well as the criticality. Thereby, embedding principle holds in a sense that loss landscape of any CNN contains all critical points of all narrower CNNs whose number of channels in each layer is no more than that of the target CNN. Currently, depth serves as a preset hyperparameter in our analysis. Whether loss landscape of DNNs of different depth has certain embedding relation for specific DNN architectures such as ResNet is an interesting open problem.

Our embedding principle and experiments in Figs. 1 and 5 suggest that whenever training of a wide DNN is stagnated around a critical point, it potentially is embedded from a much narrower DNN. Therefore, many neurons with similar representation can be reduced to one neuron. How we can design efficient pruning algorithm to fully realize this potential and how it is related to existing pruning methods as well as the well-known lottery ticket hypothesis (Frankle and Carbin, 2018) are important problems for our future research.

We remark that our embedding principle applies for landscape of general loss functions. Although for loss functions like cross entropy, a meaningful finite critical point may not exist as its parameters diverge in general throughout the training, yet it is reasonable to expect that critical embedding may provide us certain approximate critical points from narrow NNs. Of course, how to properly define an approximate critical point is in itself a problem of interest. And we leave this problem for the future study.

Overall, our embedding principle provides the first clear picture about the general structure of critical points of DNN loss landscape, which is fundamental to the theoretical understanding of both training and generalization behavior of DNNs as well as the design of optimization algorithms. Of course, the

study of loss landscape of DNN is far from complete. This work serves as a starting point for a novel line of research, which finally leads to an exact and comprehensive theoretical description about loss landscape of DNNs as well as an understanding of its profound impact on training and generalization.

## Acknowledgments and Disclosure of Funding

This work is sponsored by the National Key R&D Program of China Grant No. 2019YFA0709503 (Z. X.), the Shanghai Sailing Program, the Natural Science Foundation of Shanghai Grant No. 20ZR1429000 (Z. X.), the National Natural Science Foundation of China Grant No. 62002221 (Z. X.), the National Natural Science Foundation of China Grant No. 12101401 (T. L.), the National Natural Science Foundation of China Grant No. 12101402 (Y. Z.), Shanghai Municipal of Science and Technology Project Grant No. 20JC1419500 (Y.Z.), Shanghai Municipal of Science and Technology Major Project No. 2021SHZDZX0102, and the HPC of School of Mathematical Sciences and the Student Innovation Center at Shanghai Jiao Tong University.

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
