# Supplement to: Embedding Principle of Loss Landscape of Deep Neural Networks

Yaoyu Zhang[1,2]*, Zhongwang Zhang[1] †, Tao Luo[1], Zhi-Qin John Xu[1]‡

[1] School of Mathematical Sciences, Institute of Natural Sciences, MOE-LSC and
Qing Yuan Research Institute, Shanghai Jiao Tong University
[2] Shanghai Center for Brain Science and Brain-Inspired Technology
{zhyy.sjtu, 0123zzw666, luotao41, xuzhiqin}@sjtu.edu.cn.

## A  Appendix

**Lemma (Lemma 1).** *Given a $L$-layer ($L \geq 2$) fully-connected neural network with width $(m_0, \ldots, m_L)$, for any network parameters $\boldsymbol{\theta} = (\boldsymbol{W}^{[1]}, \boldsymbol{b}^{[1]}, \cdots, \boldsymbol{W}^{[L]}, \boldsymbol{b}^{[L]})$ and for any $l \in [L-1]$, $s \in [m_l]$, we have the expressions for $\boldsymbol{\theta}' := \mathcal{T}_{l,s}^{\alpha}(\boldsymbol{\theta})$ (see Fig. S2 for an illustration)*

*(i) feature vectors in $\boldsymbol{F}_{\boldsymbol{\theta}'}$: $\boldsymbol{f}_{\boldsymbol{\theta}'}^{[l']} = \boldsymbol{f}_{\boldsymbol{\theta}}^{[l']}$, $l' \neq l$ and $\boldsymbol{f}_{\boldsymbol{\theta}'}^{[l]} = \left[ (\boldsymbol{f}_{\boldsymbol{\theta}}^{[l]})^{\mathsf{T}}, (\boldsymbol{f}_{\boldsymbol{\theta}}^{[l]})_s \right]^{\mathsf{T}}$;*

*(ii) feature gradients in $\boldsymbol{G}_{\boldsymbol{\theta}'}$: $\boldsymbol{g}_{\boldsymbol{\theta}'}^{[l']} = \boldsymbol{g}_{\boldsymbol{\theta}}^{[l']}$, $l' \neq l$ and $\boldsymbol{g}_{\boldsymbol{\theta}'}^{[l]} = \left[ (\boldsymbol{g}_{\boldsymbol{\theta}}^{[l]})^{\mathsf{T}}, (\boldsymbol{g}_{\boldsymbol{\theta}}^{[l]})_s \right]^{\mathsf{T}}$;*

*(iii) error vectors in $\boldsymbol{Z}_{\boldsymbol{\theta}'}$: $\boldsymbol{z}_{\boldsymbol{\theta}'}^{[l']} = \boldsymbol{z}_{\boldsymbol{\theta}}^{[l']}$, $l' \neq l$*
*and $\boldsymbol{z}_{\boldsymbol{\theta}'}^{[l]} = \left[ (\boldsymbol{z}_{\boldsymbol{\theta}}^{[l]})_{[1:s-1]}^{\mathsf{T}}, (1-\alpha)(\boldsymbol{z}_{\boldsymbol{\theta}}^{[l]})_s, (\boldsymbol{z}_{\boldsymbol{\theta}}^{[l]})_{[s+1:m_l]}^{\mathsf{T}}, \alpha(\boldsymbol{z}_{\boldsymbol{\theta}}^{[l]})_s \right]^{\mathsf{T}}$.*

*Proof.* (i) By the construction of $\boldsymbol{\theta}'$, it is clear that $\boldsymbol{f}_{\boldsymbol{\theta}'}^{[l']} = \boldsymbol{f}_{\boldsymbol{\theta}}^{[l']}$ for any $l' \in [l-1]$. Then

$$\boldsymbol{f}_{\boldsymbol{\theta}'}^{[l]} = \sigma \circ \left( \begin{bmatrix} \boldsymbol{W}^{[l]} \\ \boldsymbol{W}_{s,[1:m_{l-1}]}^{[l]} \end{bmatrix} \boldsymbol{f}_{\boldsymbol{\theta}}^{[l-1]} + \begin{bmatrix} \boldsymbol{b}^{[l]} \\ \boldsymbol{b}_s^{[l]} \end{bmatrix} \right) = \begin{bmatrix} \boldsymbol{f}_{\boldsymbol{\theta}}^{[l]} \\ (\boldsymbol{f}_{\boldsymbol{\theta}}^{[l]})_s \end{bmatrix}. \tag{1}$$

Note that

$$\alpha \left[ \boldsymbol{0}_{m_{l+1} \times (s-1)}, -\boldsymbol{W}_{[1:m_{l+1}],s}^{[l+1]}, \boldsymbol{0}_{m_{l+1} \times (m_l - s)}, \boldsymbol{W}_{[1:m_{l+1}],s}^{[l+1]} \right] \begin{bmatrix} \boldsymbol{f}_{\boldsymbol{\theta}}^{[l]} \\ (\boldsymbol{f}_{\boldsymbol{\theta}}^{[l]})_s \end{bmatrix} = \boldsymbol{0}_{m_{l+1} \times 1}.$$

Thus

$$\boldsymbol{f}_{\boldsymbol{\theta}'}^{[l+1]} = \sigma \circ \left( \begin{bmatrix} \boldsymbol{W}^{[m_{l+1}]}, \boldsymbol{0}_{m_{l+1} \times 1} \end{bmatrix} \begin{bmatrix} \boldsymbol{f}_{\boldsymbol{\theta}}^{[l]} \\ (\boldsymbol{f}_{\boldsymbol{\theta}}^{[l]})_s \end{bmatrix} + \boldsymbol{0}_{m_{l+1} \times 1} + \boldsymbol{b}^{[l+1]} \right) = \boldsymbol{f}_{\boldsymbol{\theta}}^{[l+1]}. \tag{2}$$

Next, by the construction of $\boldsymbol{\theta}'$ again, it is clear that $\boldsymbol{f}_{\boldsymbol{\theta}'}^{[l']} = \boldsymbol{f}_{\boldsymbol{\theta}}^{[l']}$ for any $l' \in [l+1 : L]$.

(ii) The results for feature gradients $\boldsymbol{g}_{\boldsymbol{\theta}'}^{[l']}$, $l' \in [L]$ can be calculated in a similar way.

(iii) By the backpropagation and the above facts in (i), we have $\boldsymbol{z}_{\boldsymbol{\theta}'}^{[L]} = \nabla \ell(\boldsymbol{f}_{\boldsymbol{\theta}'}^{[L]}, \boldsymbol{y}) = \nabla \ell(\boldsymbol{f}_{\boldsymbol{\theta}}^{[L]}, \boldsymbol{y}) = \boldsymbol{z}_{\boldsymbol{\theta}}^{[L]}$.

---

*Corresponding author: zhyy.sjtu@sjtu.edu.cn.

†Part of this work is done when ZZ was an undergraduate student of Zhiyuan Honors Program at Shanghai Jiao Tong University.

‡Corresponding author: xuzhiqin@sjtu.edu.cn.

35th Conference on Neural Information Processing Systems (NeurIPS 2021).

Recalling the recurrence relation for $l' \in [l+1 : L-1]$, then we recursively obtain the following equality for $l'$ from $L-1$ down to $l+1$:

$$z_{\theta'}^{[l']} = (W^{[l'+1]})^\intercal z_{\theta'}^{[l'+1]} \circ g_{\theta'}^{[l'+1]} = (W^{[l'+1]})^\intercal z_{\theta}^{[l'+1]} \circ g_{\theta}^{[l'+1]} = z_{\theta}^{[l']}. \tag{3}$$

Next,

$$z_{\theta'}^{[l]} = \left( \left[ W^{[m_{l+1}]}, \mathbf{0}_{m_{l+1} \times 1} \right] + \alpha \left[ \mathbf{0}_{m_{l+1} \times (s-1)}, -W_{[1:m_{l+1}],s}^{[l+1]}, \mathbf{0}_{m_{l+1} \times (m_l - s)}, W_{[1:m_{l+1}],s}^{[l+1]} \right] \right)^\intercal z_{\theta}^{[l+1]} \circ g_{\theta}^{[l+1]}$$

$$= \left[ \begin{array}{c} z_{\theta}^{[l]} \\ 0 \end{array} \right] + \left[ \begin{array}{c} \mathbf{0}_{m_{l+1} \times (s-1)} \\ -\alpha(z_{\theta}^{[l]})_s \\ \mathbf{0}_{m_{l+1} \times (m_l - s)} \\ \alpha(z_{\theta}^{[l]})_s \end{array} \right]$$

$$= \left[ (z_{\theta}^{[l]})_{[1:s-1]}^\intercal, (1-\alpha)(z_{\theta}^{[l]})_s, (z_{\theta}^{[l]})_{[s+1:m_l]}^\intercal, \alpha(z_{\theta}^{[l]})_s \right]^\intercal. \tag{4}$$

Finally,

$$z_{\theta'}^{[l-1]} = \left[ (W^{[l]})^\intercal, (W^{[l]})_{s,[1:m_{l-1}]}^\intercal \right] \left( \left[ \begin{array}{c} z_{\theta}^{[l]} \\ 0 \end{array} \right] + \left[ \begin{array}{c} \mathbf{0}_{m_{l+1} \times (s-1)} \\ -\alpha(z_{\theta}^{[l]})_s \\ \mathbf{0}_{m_{l+1} \times (m_l - s)} \\ \alpha(z_{\theta}^{[l]})_s \end{array} \right] \right) \circ \left[ \begin{array}{c} g_{\theta}^{[l]} \\ (g_{\theta}^{[l]})_s \end{array} \right]$$

$$= (W^{[l]})^\intercal z_{\theta}^{[l]} \circ g_{\theta}^{[l]} + \mathbf{0}_{m_{l-1} \times 1}$$

$$= z_{\theta}^{[l-1]}. \tag{5}$$

This with the recurrence relation again leads to $z_{\theta'}^{[l']} = z_{\theta}^{[l']}$ for all $l' \in [1 : l-1]$. $\qquad\square$

**Proposition (Proposition 1: one-step embedding preserves network properties).** *Given a L-layer ($L \geq 2$) fully-connected neural network with width $(m_0, \ldots, m_L)$, for any network parameters $\theta = (W^{[1]}, b^{[1]}, \cdots, W^{[L]}, b^{[L]})$ and for any $l \in [L-1]$, $s \in [m_l]$, the following network properties are preserved for $\theta' = \mathcal{T}_{l,s}^{\alpha}(\theta)$:*

*(i) output function is preserved: $f_{\theta'}(x) = f_{\theta}(x)$ for all $x$;*

*(ii) empirical risk is preserved: $R_S(\theta') = R_S(\theta)$;*

*(iii) the sets of features are preserved:* $\left\{ \left( f_{\theta'}^{[l]} \right)_i \right\}_{i \in [m_l+1]} = \left\{ \left( f_{\theta}^{[l]} \right)_i \right\}_{i \in [m_l]}$ *and*
$\left\{ \left( f_{\theta'}^{[l']} \right)_i \right\}_{i \in [m_{l'}]} = \left\{ \left( f_{\theta}^{[l']} \right)_i \right\}_{i \in [m_{l'}]}$ *for $l' \in [L]\backslash\{l\}$;*

*Proof.* The properties (i)–(iii) are direct consequences of Lemma 1. $\qquad\square$

**Theorem (Theorem 1: criticality preserving).** *Given a L-layer ($L \geq 2$) fully-connected neural network with width $(m_0, \ldots, m_L)$, for any network parameters $\theta = (W^{[1]}, b^{[1]}, \cdots, W^{[L]}, b^{[L]})$ and for any $l \in [L-1]$, $s \in [m_l]$, if $\nabla_{\theta} R_S(\theta) = 0$, then $\nabla_{\theta} R_S(\theta') = 0$.*

*Proof.* Gradient of loss with respect to network parameters of each layer can be computed from $F$, $G$, and $Z$ as follows

$$\nabla_{W^{[l']}} R_S(\theta) = \nabla_{W^{[l']}} \mathbb{E}_S \ell(f_{\theta}(x), y) = \mathbb{E}_S \left( z_{\theta}^{[l']} \circ g_{\theta}^{[l']} (f_{\theta}^{[l'-1]})^\intercal \right),$$

$$\nabla_{b^{[l']}} R_S(\theta) = \nabla_{b^{[l']}} \mathbb{E}_S \ell(f_{\theta}(x), y) = \mathbb{E}_S (z_{\theta}^{[l']} \circ g_{\theta}^{[l']}).$$

Then, by Lemma 1, we have $\nabla_{W^{[l']}} R_S(\theta') = \nabla_{W^{[l']}} R_S(\theta) = 0$ for $l' \neq l, l+1$ and $\nabla_{b^{[l']}} R_S(\theta') = \nabla_{b^{[l']}} R_S(\theta) = 0$ for $l' \neq l$. Also, for any $j \in [m_{l+1}], k \in [m_l]$, since $(z_{\theta'}^{[l+1]})_j = (z_{\theta}^{[l+1]})_j$, $(g_{\theta'}^{[l+1]})_j = (g_{\theta}^{[l+1]})_j$, and $(f_{\theta'}^{[l]})_k = (f_{\theta}^{[l]})_k$, $(f_{\theta'}^{[l]})_{m_l+1} = (f_{\theta}^{[l]})_s$, we obtain

$$\nabla_{W_{j,k}^{[l+1]}} R_S(\theta') = \nabla_{W_{j,k}^{[l+1]}} R_S(\theta) = 0,$$

$$\nabla_{W_{j,m_l+1}^{[l+1]}} R_S(\theta') = \nabla_{W_{j,s}^{[l+1]}} R_S(\theta) = 0.$$

Similarly, for any $j \in [m_l]\backslash\{s\}, k \in [m_{l-1}]$, we have

$$\nabla_{\boldsymbol{W}^{[l]}_{j,k}} R_S(\boldsymbol{\theta}') = \nabla_{\boldsymbol{W}^{[l]}_{j,k}} R_S(\boldsymbol{\theta}) = 0,$$

$$\nabla_{\boldsymbol{b}^{[l]}_{j}} R_S(\boldsymbol{\theta}') = \nabla_{\boldsymbol{b}^{[l]}_{j}} R_S(\boldsymbol{\theta}) = 0,$$

$$\nabla_{\boldsymbol{W}^{[l]}_{s,k}} R_S(\boldsymbol{\theta}') = (1-\alpha)\nabla_{\boldsymbol{W}^{[l]}_{s,k}} R_S(\boldsymbol{\theta}) = 0,$$

$$\nabla_{\boldsymbol{W}^{[l]}_{m_l+1,k}} R_S(\boldsymbol{\theta}') = \alpha\nabla_{\boldsymbol{W}^{[l]}_{s,k}} R_S(\boldsymbol{\theta}) = 0,$$

$$\nabla_{\boldsymbol{b}^{[l]}_{s}} R_S(\boldsymbol{\theta}') = (1-\alpha)\nabla_{\boldsymbol{b}^{[l]}_{s}} R_S(\boldsymbol{\theta}) = 0,$$

$$\nabla_{\boldsymbol{b}^{[l]}_{m_l+1}} R_S(\boldsymbol{\theta}') = \alpha\nabla_{\boldsymbol{b}^{[l]}_{s}} R_S(\boldsymbol{\theta}) = 0.$$

Collecting all the above equalities, we have $\nabla_{\boldsymbol{\theta}} R_S(\boldsymbol{\theta}') = \boldsymbol{0}$. $\qquad\square$

**Lemma** (**Lemma 2: increment of the degree of degeneracy**). *Given a L-layer ($L \geq 2$) fully-connected neural network with width $(m_0, \ldots, m_L)$, if there exists $l \in [L-1]$, $s \in [m_l]$, and a q-dimensional differential manifold $\mathcal{M}$ consisting of critical points of $R_S$ such that for any $\boldsymbol{\theta} \in \mathcal{M}$, $\boldsymbol{W}^{[l+1]}_{[1:m_{l+1}],s} \neq \boldsymbol{0}$, then $\mathcal{M}' := \{\mathcal{T}^{\alpha}_{l,s}(\boldsymbol{\theta})|\boldsymbol{\theta} \in \mathcal{M}, \alpha \in \mathbb{R}\}$ is a $(d+1)$-dimensional differential manifold consists of critical points for the corresponding L-layer fully-connected neural network with width $(m_0, \ldots, m_{l-1}, m_l+1, m_{l+1}, \ldots, m_L)$.*

*Proof.* For any $\boldsymbol{\theta} \in \mathcal{M}$, let $\{e_i(\boldsymbol{\theta})\}^d_{i=1}$ be a basis of its tangent space $T_{\boldsymbol{\theta}}\mathcal{M}$. Then for any $\alpha \in \mathbb{R}$, the tangent space of $\boldsymbol{\theta}' = \mathcal{T}^{\alpha}_{l,s}(\boldsymbol{\theta}) \in \mathcal{M}'$ is spanned by $\{\mathcal{T}_{l,s}(e_1(\boldsymbol{\theta})), \cdots, \mathcal{T}_{l,s}(e_d(\boldsymbol{\theta})), \mathcal{V}_{l,s}(\boldsymbol{\theta})\}$. Since $\mathcal{T}_{l,s}$ is linear and injective, $\{\mathcal{T}_{l,s}(e_i(\boldsymbol{\theta}))\}^d_{i=1}$ is also a linearly independent set. Moreover, since $\boldsymbol{W}^{[l+1]}_{[1:m_{l+1}],m_l+1} = \boldsymbol{0}$ for any vector in parameter space applied with $\mathcal{T}_{l,s}$, then, $\{\mathcal{T}_{l,s}(e_1(\boldsymbol{\theta})), \cdots, \mathcal{T}_{l,s}(e_d(\boldsymbol{\theta})), \mathcal{V}_{l,s}(\boldsymbol{\theta})\}$ are independent if and only if $\mathcal{V}_{l,s}(\boldsymbol{\theta}) \neq \boldsymbol{0}$, i.e., $\boldsymbol{W}^{[l+1]}_{[1:m_{l+1}],s} \neq \boldsymbol{0}$. $\qquad\square$

**Remark 1.** *The requirement that $\mathcal{M}$ is a q-dimensional differential manifold can be relaxed to that $\mathcal{M}$ is a q-dimensional topological manifold. In the latter case, $\mathcal{M}'$ is a $(d+1)$-dimensional topological manifold.*

**Theorem** (**Theorem 2: degeneracy of embedded critical points**). *Consider two L-layer ($L \geq 2$) fully-connected neural networks $\text{NN}_A(\{m_l\}^L_{l=0})$ and $\text{NN}_B(\{m'_l\}^L_{l=0})$ which is K-neuron wider than $\text{NN}_A$. Suppose that the critical point $\boldsymbol{\theta}_A = (\boldsymbol{W}^{[1]}, \boldsymbol{b}^{[1]}, \cdots, \boldsymbol{W}^{[L]}, \boldsymbol{b}^{[L]})$ satisfy $\boldsymbol{W}^{[l]} \neq \boldsymbol{0}$ for each layer $l \in [L]$. Then the parameters $\boldsymbol{\theta}_A$ of $\text{NN}_A$ can be critically embedded to a K-dimensional critical affine subspace $\mathcal{M}_B = \{\boldsymbol{\theta}_B + \sum^K_{i=1} \alpha_i v_i|\alpha_i \in \mathbb{R}\}$ of loss landscape of $\text{NN}_B$. Here $\boldsymbol{\theta}_B = (\prod^K_{i=1} \mathcal{T}_{l_i,s_i})(\boldsymbol{\theta}_A)$ and $v_i = \mathcal{T}_{l_K,s_K} \cdots \mathcal{V}_{l_i,s_i} \cdots \mathcal{T}_{l_1,s_1} \boldsymbol{\theta}_A$.*

*Proof.* The assumption $\boldsymbol{W}^{[l]} \neq \boldsymbol{0}$, $l \in [L]$ implies the existence of non-silent neurons, i.e., existing $s \in [m_l]$ such that $\boldsymbol{W}^{[l+1]}_{[1:m_{l+1}],s} \neq \boldsymbol{0}$, for any $l \in [L-1]$ with $m'_l > m_l$.

In this proof, we misuse notation and denote $m_l = m_l(\boldsymbol{\theta})$ for the width of the $l$-th layer for any fully-connected neural network with parameters $\boldsymbol{\theta}$. For such a general network with parameters $\boldsymbol{\theta}$, we introduce the following operator. Given an index set $J$ and for any $l \in [L]$, $s \in [m_l]$, we define

$$\mathcal{V}_{l,s,J}(\boldsymbol{\theta}) = \left(\boldsymbol{0}_{m_0 \times m_1}, \cdots, \left[\boldsymbol{0}_{m_{l+1} \times (s-1)}, -\sum_{j \in J} \boldsymbol{W}^{[l+1]}_{[1:m_{l+1}],j}, \boldsymbol{0}_{m_{l+1} \times (m_l-s)}, \sum_{j \in J} \boldsymbol{W}^{[l+1]}_{[1:m_{l+1}],j}\right], \cdots\right).$$

Clearly, $\mathcal{V}_{l,s,J} = \sum_{j \in J} \mathcal{V}_{l,s,\{j\}}$. If for all $j \in J$, $\boldsymbol{W}^{[l]}_{j,[1:m_{l-1}]} = \boldsymbol{W}^{[l]}_{s,[1:m_{l-1}]}$, $\boldsymbol{W}^{[l+1]}_{[1:m_{l+1}],j} = \beta_j \sum_{j' \in J} \boldsymbol{W}^{[l+1]}_{[1:m_{l+1}],j'}$ and $\sum_{j' \in J} \boldsymbol{W}^{[l+1]}_{[1:m_{l+1}],j'} \neq \boldsymbol{0}$, then for $\beta_s \neq 0$, we have

$$\mathcal{T}^\alpha_{l,s,J}\boldsymbol{\theta} = (\mathcal{T}_{l,s} + \alpha\mathcal{V}_{l,s,J})\boldsymbol{\theta}$$
$$= (\mathcal{T}_{l,s} + \alpha \sum_{j \in J} \mathcal{V}_{l,s,j})\boldsymbol{\theta}$$
$$= (\mathcal{T}_{l,s} + \alpha \sum_{j \in J} \frac{\beta_j}{\beta_s}\mathcal{V}_{l,s})\boldsymbol{\theta}$$
$$= \mathcal{T}^{\alpha'}_{l,s}\boldsymbol{\theta},$$

where $\alpha' = \alpha \sum_{j \in J} \frac{\beta_j}{1-\beta_j}$, is simply the one-step critical embedding. We can extend this to the case of $\beta_s = 0$.

Then, for $J' = J \cup \{m_l + 1\}$, $l' = l$, $s' \in J$,

$$\mathcal{V}_{l',s',J'}\mathcal{V}_{l,s,J} = (\mathcal{V}_{l,s',J} + \mathcal{V}_{l,s',m_l+1})\mathcal{V}_{l,s,J}$$
$$= \mathcal{V}_{l,s',J}\mathcal{V}_{l,s,J} + \mathcal{V}_{l,s',m_l+1}\mathcal{V}_{l,s,J}$$
$$= \mathcal{V}_{l,s',s}\mathcal{V}_{l,s,J} + \mathcal{V}_{l,s',m_l+1}\mathcal{V}_{l,s,J}$$
$$= \boldsymbol{0}.$$

In general, we have

$$\mathcal{V}_{l',s',J'} \prod_{i=1}^N \mathcal{T}_{l'_i,s'_i}\mathcal{V}_{l,s,J} = (\mathcal{V}_{l,s',J} + \mathcal{V}_{l,s',m_l+1}) \prod_{i=1}^N \mathcal{T}_{l'_i,s'_i}\mathcal{V}_{l,s,J}$$
$$= \mathcal{V}_{l,s',J} \prod_{i=1}^N \mathcal{T}_{l'_i,s'_i}\mathcal{V}_{l,s,J} + \mathcal{V}_{l,s',m_l+1} \prod_{i=1}^N \mathcal{T}_{l'_i,s'_i}\mathcal{V}_{l,s,J}$$
$$= \mathcal{V}_{l,s',s} \prod_{i=1}^N \mathcal{T}_{l'_i,s'_i}\mathcal{V}_{l,s,J} + \mathcal{V}_{l,s',m_l+1} \prod_{i=1}^N \mathcal{T}_{l'_i,s'_i}\mathcal{V}_{l,s,J}$$
$$= \boldsymbol{0}.$$

For $l' \neq l$ or $s' \notin J$, obviously we have $\mathcal{V}_{l',s',J'}\mathcal{V}_{l,s,J} = \boldsymbol{0}$ and $\mathcal{V}_{l',s',J'} \prod_{i=1}^N \mathcal{T}_{l'_i,s'_i}\mathcal{V}_{l,s,J} = \boldsymbol{0}$.

Now we are ready to prove the lemma. Let $J_i = \{s_i\} \cup \{m_l + \#\{i|l_i = l, i \in [j]\}|l_j = l, s_j = s, j \in [K]\}$, where $\#$ indicates number of elements in a set. Then

$$\prod_{i=1}^K \mathcal{T}^{\alpha_i}_{l_i,s_i,J_i} = \prod_{i=1}^K (\mathcal{T}_{l_i,s_i} + \alpha_i\mathcal{V}_{l_i,s_i,J_i})$$
$$= \prod_{i=1}^K \mathcal{T}_{l_i,s_i} + \sum_{i=1}^K \alpha_i\mathcal{T}_{l_K,s_K} \cdots \mathcal{V}_{l_i,s_i,J_i} \cdots \mathcal{T}_{l_1,s_1},$$
$$= \prod_{i=1}^K \mathcal{T}_{l_i,s_i} + \sum_{i=1}^K \alpha_i\mathcal{T}_{l_K,s_K} \cdots \mathcal{V}_{l_i,s_i} \cdots \mathcal{T}_{l_1,s_1},$$

which is a critical embedding for any $[\alpha_i]^K_{i=1} \in \mathbb{R}^K$. This completes the proof.

$\square$

## A.1 Trivial critical transforms

In general, neuron-index permutation among the same layer is a trivial criticality invariant transform because of the layer-wise intrinsic symmetry of DNN models. Therefore, any critical point/manifold

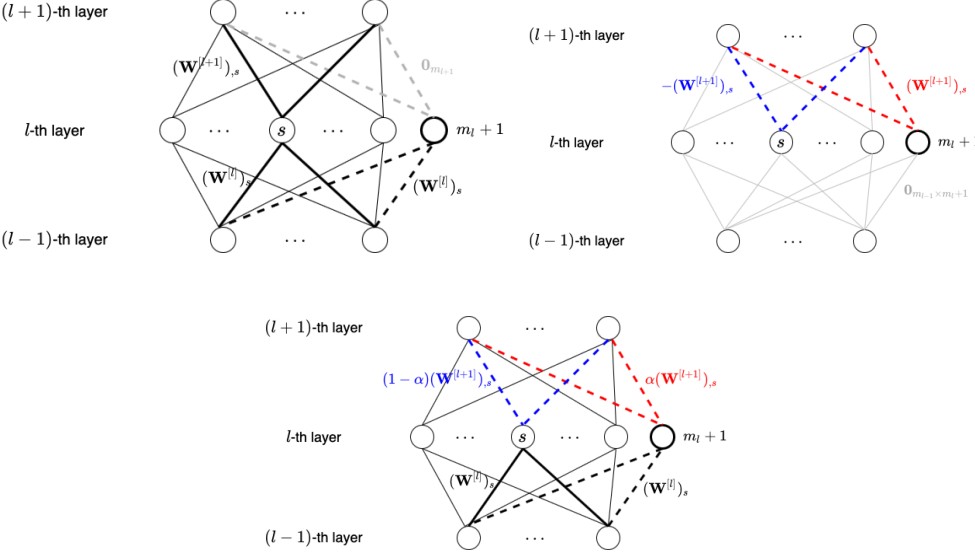

Figure S1: Illustration of $\mathcal{T}_{l,s}$, $\mathcal{V}_{l,s}$, and $\mathcal{T}_{l,s}^{\alpha}$.

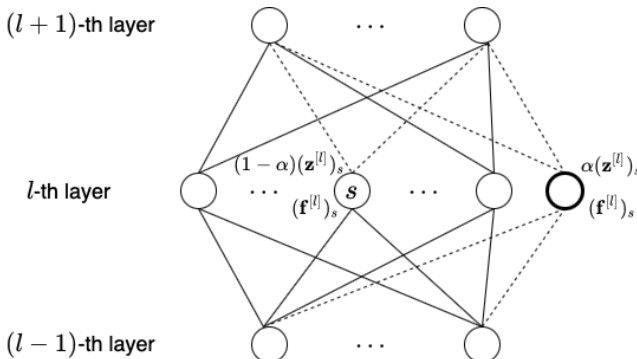

Figure S2: Illustration of $\boldsymbol{F}$ and $\boldsymbol{Z}$

may result in multiple "mirror" critical points/manifolds of the loss landscape through all possible permutations. However, this transform does not inform about the degeneracy of critical points/manifolds.

For any $p$-homogeneous activation function $\sigma$ i.e., $\sigma(\beta x) = \beta^p \sigma(x)$ for any $\beta > 0$ and $x \in \mathbb{R}$, we define for any $l \in [L-1]$, $s \in [m_l]$ the following scaling transform $\boldsymbol{\theta}' = \mathcal{S}_{l,s}^{\beta}(\boldsymbol{\theta})$ ($\beta \neq 0$) such that $\boldsymbol{W}_{[1:m_{l+1}],s}'^{[l+1]} = \frac{1}{\beta^p} \boldsymbol{W}_{[1:m_{l+1}],s}^{[l+1]}$ and $\boldsymbol{W}_{s,[1:m_{l-1}]}'^{[l]} = \beta \boldsymbol{W}_{s,[1:m_{l-1}]}^{[l]}, \boldsymbol{b}_s'^{[l]} = \beta \boldsymbol{b}_s^{[l]}$, and all the other entries remain the same. Clearly, this transform is also a critical transform. Moreover, it informs about one more degenerate dimension for each neuron with $\left\| \boldsymbol{W}_{[1:m_{l+1}],s}^{[l+1]} \right\|_2 \left\| \left( \boldsymbol{W}_{s,[1:m_{l-1}]}^{[l]\mathsf{T}}, \boldsymbol{b}_s^{[l]} \right)^{\mathsf{T}} \right\|_2 \neq 0$. This critical scaling transform is trivial in a sense that it is an obvious result of the cross-layer scaling preserving intrinsic to each DNN of homogeneous activation function, not relevant to cross-width landscape similarity between DNNs we focus on.

# B   Details of experiments

For the 1D fitting experiments (Figs. 1, 3(a), 4), we use tanh as the activation function, mean squared error (MSE) as the loss function. We use the full-batch gradient descent with learning rate 0.005 to train NNs for 300000 epochs. The initial distribution of all parameters follows a normal distribution with a mean of 0 and a variance of $\frac{1}{m^3}$.

For the iris classification experiment (Fig. 3(b)), we use sigmoid as the activation function, MSE as the loss function. We use the default Adam optimizer of full batch with learning rate 0.02 to train for 500000 epochs. The initial distribution of all parameters follows a normal distribution with mean 0 and variance $\frac{1}{m^6}$.

For the experiment of MNIST classification (Fig. 5), we use ReLU as the activation function, MSE as the loss function. We also use the default Adam optimizer of full batch with learning rate 0.00003 to train for 100000 epochs. The initial distribution of all parameters follows a normal distribution with mean 0 and variance $\frac{1}{m^6}$.

To obtain the empirical diagram in Fig. 4, we run 200 trials each for width-1, width-2 and width-3 tanh NNs with variance of initial parameters $\frac{1}{m^3}$ ($m = 1, 2, 3$) for 300000 epochs. Then we find all parameters with gradient less than $10^{-10}$, which we define as empirical critical points, throughout the training in total 600 trajectories. Next, we cluster them based on their loss values, output functions, input parameters of neurons and only 4 different cases arises after excluding the trivial case of constant zero output. Their output functions are shown in the figure.

Remark that, although Figs. 1 and 5 are case studies each based on a random trial, similar phenomenon can be easily observed as long as the initialization variance is properly small, i.e., far from the linear/kernel/NTK regime.