# OpenReview forum: "Embedding Principle of Loss Landscape of Deep Neural Networks"
_NeurIPS.cc/2021/Conference — NeurIPS 2021 Spotlight_

### Official Review · Reviewer_FvEo · 2021-07-15

**Rating:** 7
**Confidence:** 3

**Summary:**

The paper discusses the question of organization of loss surfaces of neural networks, in particular the connection between wider and narrower networks with respect to the critical points. The proposed analysis introduces an embedding that projects a narrower network into a wider network (at least by one neuron in at least one hidden layer) without changing its function (output) and also saving all its critical points. It is shown, that the introduced procedure of the mapping narrower networks into wider increases degeneracy of the critical points, i.e., increasing amount of zero valued eigenvalues.

**Limitations And Societal Impact:**

The paper discusses limitations; as a theoretical paper it does not have a direct societal impact.

**Main Review:**

The overall work seems very interesting and shedding light on the organization of a loss surfaces of neural networks. Interesting future directions are discussed in section6, including the open question with depth (how depth affects the critical points of the loss surface) as well as identification of critical points.

There are still some points that raise questions, in particular:
- It is never discussed if the theory is completely independent of the type of loss function and the activation functions.
- I think the discussion about the degeneracy of the embedded critical points can be improved: in particular the remark in 3.1 does not seem straightforward conclusion from the informal Theorem2, while theorems in section5 are not discussed anymore.
- Possibly some definitions are missing (e.g., critical hyperplane used in informal statement of theorem2).
- I found hard to follow the diagram in Figure4 - how exactly it can be claimed that all the possible critical points for a neural network are found?
- It is hard to understand the provided numerical experiments, in particular because the type of neural networks is not described; the definition of "training till critical point" is not given (I would suggest that the gradients are checked to be 0, but judging by the provided code it is very large amount of epochs). The provided code is not helpful - the images are build based on one of the random runs of the scripts and it is unclear how common is the observed picture.

The contribution of the paper is very interesting, but the introduced numerical experiments are raising more questions than contribute to the paper---this should be improved.

-----
After the rebuttal and editions I am raising the score to accept (7).

**Time Spent Reviewing:**

4

---

> ### Author Response · Authors · 2021-08-09
> **Response to Reviewer FvEo**
>
> $\textbf{Point 1:}$
> It is never discussed if the theory is completely independent of the type of loss function and the activation functions.
>
> $\textbf{Reply:}$
>
> We add assumptions required for all our results in Section 3.1 to clarify this issue.
>
> "Assumptions:
>
> (i) $L$-layer ($L\geq 2$) fully-connected NN.
>
> (ii) Training data $S=\{(x_i,y_i)\}_{i=1}^n$, $n\in\mathbb{Z}^+\cup\{+\infty\}$.
>
> (iii) $R_S(\theta)=E_S \ell (f_{\theta}(x),y)$ without explicit regularization term.
>
> (iv) Loss and activation functions are differentiable. Note that, even for functions like ReLU or hinge loss, as long as we uniquely assign a subgradient to their non-differentiable points, all our results still hold."
>
> $\textbf{Point 2: }$
> I think the discussion about the degeneracy of the embedded critical points can be improved: in particular the remark in 3.1 does not seem straightforward conclusion from the informal Theorem2, while theorems in section5 are not discussed anymore.
>
> $\textbf{Point 3:}$
> Possibly some definitions are missing (e.g., critical hyperplane used in informal statement of theorem2).
>
> $\textbf{Reply for Point 2 and 3:}$
>
> As suggested, we add the following definition about the critical submanifold/affine subspace in Section 3.1. We change all "critical hyperplane" to "critical affine subspace" throughout the paper to avoid confusion.
>
> "Definition: (critical submanifold/affine subspace).
>
> A critical submanifold or affine subspace $\mathcal{M}$ is a connected subsubmanifold or affine subspace of the parameter space $\mathbb{R}^M$, such that each $\theta\in\mathcal{M}$ is a critical point of loss with the same loss value."
>
> We also add a remark later in the same section.
>
> "Remark:
>
>  In the above definition of degree of degeneracy, we require twice differentiable activation function and twice differentiable loss to compute Hessian for convenience. For loss and activation functions with only first-order differentiability, we extend the definition of degree of degeneracy as follows: for any critical point $\theta$ belonging to a $K$-dimensional critical submanifold $\mathcal{M}$, its degree of degeneracy is at least $K$."
>
> $\textbf{Point 4:}$
> I found hard to follow the diagram in Figure4 - how exactly it can be claimed that all the possible critical points for a neural network are found?
>
> $\textbf{Reply:}$
>
> For clarification, we revise the "diagram of loss landscape" to "empirical diagram of loss landscape" as"Empirical diagram of loss landscape of a width-$3$ two-layer tanh NN, i.e., all critical points that width-3 or narrower NNs may get close to during the training under proper initialization." in the caption.
>
> We also add the following description of the detailed experimental procedure in Appendix B.
>
> "To obtain the empirical diagram in Fig. 4, we run 200 trials each for width-1, width-2 and width-3 tanh NNs with variance of initial parameters $\frac{1}{m^3}$ ($m=1,2,3$) for $300000$ epochs. Then we find all parameters with gradient less than $10^{-10}$, which we define as empirical critical points, throughout the training in total $600$ trajectories. Next, we cluster them based on their loss values, output functions, input parameters of neurons and only $4$ different cases arises after excluding the trivial case of constant zero output. Their output functions are shown in the figure. "
>
> $\textbf{Point 5:}$
> It is hard to understand the provided numerical experiments, in particular because the type of neural networks is not described; the definition of "training till critical point" is not given (I would suggest that the gradients are checked to be 0, but judging by the provided code it is very large amount of epochs). The provided code is not helpful - the images are build based on one of the random runs of the scripts and it is unclear how common is the observed picture.
>
> $\textbf{Reply:}$
>
> In our work, we consider one type of network structure, that is, fully-connected networks, and we add the details of experiments in section 3.2 as follows,
>
> "Throughout this work, we use two-layer fully-connected neural network with size $d$-$m$-$d_{out}$. The input dimension $d$ is determined by the training data. The output dimension $d_{out}$ is different for different experiments. The number of hidden neurons $m$ is specified in each experiment. All parameters are initialized by a Gaussian distribution with mean zero and variance specified in each experiment. We use MSE loss trained by full batch gradient descent for 1D fitting problems (Figs. 1, 3(a) and 4), and default Adam optimizer with full batch for others. The learning rate is fixed throughout the training. More details of experiments are shown in Appendix B.
> "
>
> As for the definition of "training till critical point", we clarify it in the revised manuscript in Section 3.2 as follows,
>
> "We first roughly estimate the possible interval of critical points by observing where the loss decays very slowly, and then take the point with the smallest derivative of the parameters (use $L_1$ norm) as an empirical critical point. The $L_1$ norm of the derivative of loss function at the empirical critical point is approximately $7.15\times 10^{-15}$ for Fig.3(a) and $3.72\times 10^{-13}$ for Fig.3(b), which are reasonably small."
>
> In Figs. 1 and 5, loss trajectory for a random trial each is shown. We actually have run many trials, and tried different initialization variances. We observe that as long as the initialization variance is properly small, stagnation of loss during the training is easy to observe, and the structure of NN parameters at stagnation are qualitatively similar to the cases shown here. We add the following remark at the end of Appendix B.
>
> "Remark that, although Figs. 1 and 5 are case studies each based on a random trial, similar phenomenon can be easily observed as long as the initialization variance is properly small, i.e., far from the linear/kernel/NTK regime."
>
> The code is also updated with more comments.

---

> > ### Comment · Reviewer_FvEo · 2021-08-14
> > **Feedback**
> >
> > I thank the authors for their explanations and clarifications.
> >
> > I am quite happy with all the editions and explanations. I have one additional question now though: Why it is required to have full batch training? Is it anyhow limits the empirical evaluation? Can the same results be obtained with mini-batch or SGD training?

---

> > > ### Author Response · Authors · 2021-08-21
> > > **Response to Reviewer FvEo**
> > >
> > > $\textbf{Point 1:}$
> > > Why it is required to have full batch training? Is it anyhow limits the empirical evaluation? Can the same results be obtained with mini-batch or SGD? training?
> > >
> > > $\textbf{Reply:}$
> > >
> > > In our experiments, we use full-batch training simply as a convenient default setting. In fact, our empirical results hold for mini-batch or SGD training. For demonstration, we repeated the experiment in Fig. 5 using SGD with batch-size 32 and found similar results (details can be found on https://www.dropbox.com/s/6km9v74vptg7iz4/2343_response.pdf?dl=0).

---

### Official Review · Reviewer_TiiQ · 2021-07-16

**Rating:** 7
**Confidence:** 4

**Summary:**

This paper studies critical points of loss landscape of deep neural networks. The authors prove that all critical points of a narrow fully-connected network can be embedded to critical points of a wider network. Moreover, the degree of degeneracy of critical points of narrow networks increases under the described embedding. The critical embedding principle provides a new explanation of the existence of high-dimensional manifolds of critical points in the loss landscape of wide fully connected networks.

The embedding of a weight vector of a narrow DNN into the weight space of a wider DNN is defined as a composition of simple one-step embedding steps: at each step one neuron of a network is split into two with a specifically chosen assignment of input/output weights to both old and new neurons. The input weights for both new neurons are copied from the original neuron and output weights are re-parameterized via linear interpolation. The authors show that this one-step embedding preserves the output function of the network, criticality of a weight vector in the loss landscape, and increases the degree of degeneracy of the critical point.

The authors support the theoretical results with experiments on simple learning tasks (1D regression, classification on Iris and MNIST dataset). The experiments confirm the theoretical results: the embedding procedure preserves criticality and increases degeneracy.


**Limitations And Societal Impact:**

I think the authors adequately addressed the limitations and societal impact of their work. I do not foresee any direct negative societal impact of this paper.

**Main Review:**

To the best of my knowledge the results presented in this paper are novel. While it is known from prior work that loss landscapes of neural networks have high-dimensional manifolds of critical points, this paper identifies one particular mechanism by which these manifolds are generated.

The theoretical results of the paper are clearly formulated and technically sound. The proofs are provided and correct (to the best of my understanding). The described experimental setups and results are adequate.

In my opinion, the main strength of the paper is that it identifies an interesting structure of critical points in the loss landscapes of neural networks. The further development of the framework described in this paper can help to identify new regular structures in the loss landscape and provide novel intuition on the optimization problems in deep learning.

I have a few concerns about the paper:

1) In the abstract (line 7), the authors claim that “the embedding structure of critical points is independent of loss function and training data”. Also, in Section 3.1 (line 129) it is stated that “the degeneracy of critical points is actually a key characteristic of the loss landscape of NNs independent of the size of training data”. These claims are not supported by formal arguments and I am not sure whether the claims are valid. If we increase the number of data points the loss landscape will change and points which were critical before, might no longer be critical (both for narrow and wide networks). All theoretical results in the paper assume that point \theta_{narr} is a critical point of the current loss landscape. It is not clear to me how the structure of the critical points and the degrees of degeneracy can be independent of training data.

2) In my opinion, the paper overemphasizes the importance of the embedding principle for explanation of phenomena observed in practice and the significance of the results seems limited to me. I agree with the authors that the embedding principle provides a new perspective on the structure of the critical points and the principle suggests that the training might be implicitly regularized towards functions represented by narrow neural networks. However, there is no clear demonstration of these regularization mechanisms in action and their connection to the embedding principle is vague. I think that the paper does not convincingly show that “the embedding principle provides an explanation for the general easy optimization” (abstract, line 11). I believe that more detailed experimental investigation is needed to demonstrate how the existence of embedded critical points in the loss landscape influences training.

3) The embedding principle is based on a particular operation which increases the width of the network in such a way that the new wider network implements the same input-to-output mapping as the original network. This idea is closely related to model compression, network pruning and recent work on the lottery ticket hypothesis [1]. Unfortunately, this body of prior work is overlooked in the paper. I would encourage the authors to include the overview of these research areas and discuss the relation between the embedding principle and network pruning ideas.


Other comments:
* Line 170. Notation a_k, w_k is not defined.
* Some details of experiments (used batch sizes, optimizers, learning rate schedules) are not provided in the text of the paper.
* Shouldn’t the statement of Theorem 2 (Section 5) require that \theta_A is a critical point? What does it mean (formally) that “$\theta_A$ can be critically embedded to a critical hyperplane?”
* The description of one step embedding procedure is presented for a neuron of a fully-connected layer. Does the same procedure generalize to convolutional layers? It would be helpful to include the discussion of this topic, since the description of the convolutional layers in terms of individual neurons is not trivial.


References:

[1] Jonathan Frankle, Michael Carbin, “The Lottery Ticket Hypothesis: Finding Sparse, Trainable Neural Networks”, ICLR 2019


====================================================================

Post rebuttal update

====================================================================


After reading the author's response, I am satisfied with the revised version of the paper. I recommend the paper for acceptance. I change my original score from 6 (marginally above the acceptance threshold) to 7 (good paper accept).


**Time Spent Reviewing:**

7

---

> ### Author Response · Authors · 2021-08-09
> **Response to Reviewer TiiQ**
>
> $\textbf{Point 1:}$
> "the embedding structure of critical points is independent of loss function and training data"
>
> $\textbf{Reply:}$
>
> We revise these sentences as follows to clarify our argument:
>
>
> 1. "Note that, given any training data, differentiable loss function and differentiable activation function, this embedding structure of critical points holds.”
>
> 2. "However, we demonstrate by the above theorem that regardless of whether the NN is over-parameterized, degenerate critical points are prevalent in its loss landscape as long as narrower DNNs possess critical points."
>
> As pointed out by the reviewer that the loss landscape and its critical points depend on many factors like training data, loss function, activation function and depth. And their dependence is often dauntingly complicated for analysis. However, as demonstrated in our work, there is a simple embedding relation in width that holds regardless of what training data, differentiable loss, differentiable activation function, and depth are used. This generality is the strength of our embedding principle.
>
> $\textbf{Point 2:}$
> In my opinion, the paper overemphasizes the importance of the embedding principle for explanation of phenomena observed in practice and the significance of the results seems limited to me. I agree with the authors that the embedding principle provides a new perspective on the structure of the critical points and the principle suggests that the training might be implicitly regularized towards functions represented by narrow neural networks. However, there is no clear demonstration of these regularization mechanisms in action and their connection to the embedding principle is vague. I think that the paper does not convincingly show that "the embedding principle provides an explanation for the general easy optimization" (abstract, line 11). I believe that more detailed experimental investigation is needed to demonstrate how the existence of embedded critical points in the loss landscape influences training.
>
> $\textbf{Reply:}$
>
> We thank the reviewer for pointing out the inaccuracy of our statement and hence revise as follows :
>
> "The embedding principle provides a new perspective to study the general easy optimization of wide DNNs and …" (line 11).
>
> The optimization property of DNN training process and the implicit regularization are very important and difficult problems. Currently, we are still far from obtaining decisive results on these problems. However, the embedding principle and our case studies at a primitive stage sparkle new thoughts.
>
> Our embedding principle suggests that, comparing with studying about whether "bad" critical points exist for certain fixed width DNN, it is more meaningful to figure out whether a "bad" critical point (if exists), say a local minimum,  may steadily improve, e.g., gaining negative eigenvalues in Hessian as in Fig. 3, through each step of embedding. Or we can ask how difficult it is to have a truly bad critical point which never becomes a strict saddle point through any critical embedding.
>
> About implicit regularization, our work suggests that, comparing to estimating the complexity of DNN based on its actually number of parameters, it is more meaningful to study its effective number of parameters at critical points it may converge to, which may be far less and thus no need to worry about overfitting as exemplified in Fig. 1.
>
> A more detailed and systematic experimental investigation as well as theoretical justification on these problems, important in their own right, will be studied in our future work.
>
> $\textbf{Point 3:}$
>  I would encourage the authors to include the overview of these research areas and discuss the relation between the embedding principle and network pruning ideas.
>
> $\textbf{Reply:}$
>
> We add the following discussion about the relation between embedding principle and pruning in Section 6:
>
> "Our embedding principle and experiments in Figs. 1 and 5 suggest that whenever training of a wide DNN is stagnated around a critical point, it potentially is embedded from a much narrower DNN. Therefore, many neurons with similar representation can be reduced to one neuron. How we can design efficient pruning algorithm to fully realize this potential and how it is related to existing pruning methods as well as the well-known lottery ticket hypothesis (Frankle et al., 2019) are important problems for our future research."
>
> Embedding principle predicts a special structure of parameters of wide DNNs that has clear pruning potential. However, grouping neurons with a similar response function together efficiently is not easy in deep networks, and we do not see any pruning method that directly employs this structure.
>
> $\textbf{Point 4:}$
> Line 170. Notation $a_k$, $w_k$ is not defined.
>
> $\textbf{Reply:}$
>
> We add the definition as "…two-layer ReLU NN $f_{\theta}=\sum_{k=1}^m a_k\sigma(w_k^T\tilde{x})$ ($\tilde{x} = [x^T,1]^T$) …"
>
> $\textbf{Point 5:}$
> Some details of experiments (used batch sizes, optimizers, learning rate schedules) are not provided in the text of the paper.
>
> $\textbf{Reply:}$
>
> We add in the section 3.2 the following details of experiments.
>
> "Throughout this work, we use two-layer fully-connected neural network with size $d$-$m$-$d_{out}$. The input dimension $d$ is determined by the training data. The output dimension $d_{out}$ is different for different experiments. The number of hidden neurons $m$ is specified in each experiment. All parameters are initialized by a Gaussian distribution with mean zero and variance specified in each experiment. We use MSE loss trained by full batch gradient descent for 1D fitting problems (Figs. 1, 3(a) and 4), and default Adam optimizer with full batch for others. The learning rate is fixed throughout the training. More details of experiments are shown in Appendix B.
> "
>
> We add in the appendix the following details of experiments.
>
> "For the 1D fitting experiments (Figs. 1, 3(a) and 4), we use tanh as the activation function, mean squared error (MSE) as the loss function. We use the full-batch gradient descent with learning rate 0.005 to train NNs for 300000 epochs. The initial distribution of all parameters follows a normal distribution with a mean of 0 and a variance of $\frac{1}{m^3}$.
>
> For the iris classification experiment (Fig. 3(b)), we use sigmoid as the activation function, MSE as the loss function. We use the default Adam optimizer of full batch with learning rate 0.02 to train for 500000 epochs. The initial distribution of all parameters follows a normal distribution with mean $0$ and variance $\frac{1}{m^6}$.
>
> For the experiment of MNIST classification (Fig. 5), we use ReLU as the activation function, MSE as the loss function. We also use the default Adam optimizer of full batch with learning rate 0.00003 to train for 100000 epochs. The initial distribution of all parameters follows a normal distribution with mean $0$ and variance $\frac{1}{m^6}$."
>
> $\textbf{Point 6:}$
> Shouldn’t the statement of Theorem 2 (Section 5) require that $\theta_A$ is a critical point? What does it mean (formally) that "$\theta_A$ can be critically embedded to a critical hyperplane?"
>
> $\textbf{Reply:}$
>
> We thank the reviewer for pointing this out. Theorem 2 (Section 5) does require that $\theta_A$ is a critical point. We add this condition in the statement of theorem "…Suppose that the critical point $\theta_A$ …"
>
> For the second issue, we add the following definition about the critical submanifold/affine subspace in Section 3.1.Note that, to avoid confusion, we change all "critical hyperplane" to "critical affine subspace" throughout the paper.
>
> "Definition: (critical submanifold/affine subspace).
>
> A critical submanifold or affine subspace $\mathcal{M}$ is a connected subsubmanifold or affine subspace of the parameter space $\mathbb{R}^M$, such that each $\theta\in\mathcal{M}$ is a critical point of loss with the same loss value."
>
> $\textbf{Point 7:}$
> The description of one step embedding procedure is presented for a neuron of a fully-connected layer. Does the same procedure generalize to convolutional layers? It would be helpful to include the discussion of this topic, since the description of the convolutional layers in terms of individual neurons is not trivial.
>
> $\textbf{Reply:}$
>
> We add the following discussion in Section 6:
>
> "Considering convolutional neural networks for example, the quantity that corresponds to width of fully-connected NNs is channel. Similar to one-step or multi-step embedding, we can introduce a feature splitting operation, i.e., increase the number of channels by splitting all neurons sharing one convolution kernel with the same $\alpha$, which can be proven to preserve the output function, representation as well as the criticality. Thereby, embedding principle holds in a sense that loss landscape of any CNN contains all critical points of all narrower CNNs whose number of channels in each layer is no more than that of the target CNN. "

---

> > ### Comment · Reviewer_TiiQ · 2021-08-22
> > **Feedback**
> >
> > I read through the authors’ response as well as other reviews and corresponding responses.
> >
> > I want to thank the authors for a comprehensive response to my questions. The edits made by the authors improved the clarity of the paper, provided important details of theoretical and empirical contributions, and added reference to prior work in related research areas.
> >
> > After reading the author's response, I am satisfied with the revised version of the paper. I recommend the paper for acceptance. I change my original score from 6 (marginally above the acceptance threshold) to 7 (good paper accept).

---

### Official Review · Reviewer_Vyz8 · 2021-07-18

**Rating:** 6
**Confidence:** 2

**Summary:**

The authors introduce a prove, in case of fixed-depth fully-connected networks, what they call an "embedding principle", stating that a network who's layers are wider than layers of narrower network contain all critical points of the narrower network.

**Limitations And Societal Impact:**

Yes

**Main Review:**

Overall I like the paper and think it's an important contribution. I am not an expert on the subfield so it is possible that I missed some papers and cannot therefore vouch for its originality, but on the face value the paper addresses an important aspect of loss landscape analysis, delivers a theoretical results, and verifies it using empirical experiments.

A few papers that I thought might be relevant:
https://arxiv.org/abs/2010.15110 on the kernel and linear learning mentioned on line 268

I am giving a weak accept since I'm not an expert on the specific subdomain but I'm happy to increase my score after discussing this with other reviewers and the AC + after the author' rebuttal.

**Time Spent Reviewing:**

2

---

> ### Author Response · Authors · 2021-08-09
> **Response to Reviewer Vyz8**
>
> $\textbf{Point 1:}$
> A few papers that I thought might be relevant: https://arxiv.org/abs/2010.15110 on the kernel and linear learning mentioned on line 268.
>
> $\textbf{Reply:}$
>
> We would like to thank Reviewer’s support and constructive comments.
> We have added several literatures that study in detail the training of DNN beyond linear/kernel/NTK regime including Fort et al., (2020) as mentioned by reviewer, i.e.,
> "This may be the reason underlying the general easy optimization of wide DNNs observed in practice even beyond the linear/kernel/NTK regime (Williams et al. 2019; Fort et al., 2020; Geiger et al., 2020; Chen et al., 2020; Luo et al., 2021)."

---

> > ### Comment · Reviewer_Vyz8 · 2021-08-31
> > **A response to a response**
> >
> > I would like to thank the authors for responding to my reply. As mentioned, I am not an expert on the subfield, but reading the author reviewers and authors' responses I would like to maintain my score of 6, leaning towards accepting if I am a vote on the margin. Thank you!

---

### Official Review · Reviewer_TrtW · 2021-07-22

**Rating:** 7
**Confidence:** 3

**Summary:**

The paper proposes and proves an embedding principle that critical points of all narrow DNNs are embedded into the loss landscape of a wider DNN.

**Limitations And Societal Impact:**

Authors have detailed a number of future directions, which address some of the limitations of the current work.

**Main Review:**

The paper opens up a fundamentally novel direction for understanding loss landscape of DNNs with implication to DNN training, generalization and robustness. The work provides a principled and elegant framework for characterizing loss landscapes of DNNs, which  connects DNN capacity with its generalization ability. Results of numerical experiments on three different datasets (synthetic, Iris, and Mnist) demonstrate relevance of the proposed embedding in DNN training. For detailed comments, please see below.

(1) The details of how critical points of a narrow network are embedded to a wider DNN in numerical experiments are not provided in the paper. I would suggest authors to elaborate on this point.
(2) Authors should add DNN training details (optimizer used, learning rate, number of epochs, etc) in the paper.
(3) The paper is nicely written with a comprehensive related work review, though difficult to follow at certain sections.  I would suggest authors to revise the writing fro better readability and ease of following. Also, some of the figure captions can be improved for better reader comprehension, for example the caption of Figure 4.

**Time Spent Reviewing:**

2

---

> ### Author Response · Authors · 2021-08-09
> **Response to Reviewer TrtW**
>
> $\textbf{Point 1:}$
> The details of how critical points of a narrow network are embedded to a wider DNN in numerical experiments are not provided in the paper. I would suggest authors to elaborate on this point.
>
> $\textbf{Reply}$
>
> We add in Fig.3's caption the following details of experiments.
>
> "We equally split one neuron of a width-2 two-layer NN at a critical point into k neurons (k=2 or 3), whose input weights remain the same but output weights are 1/k of the original neuron."
>
> We add in Fig.4's caption the following details of experiments.
>
> "We use the same equal splitting as Fig. 3 to embed critical points of width-1 or width-2 NN to critical points of the width-3 NN and compute the hessian to obtain the corresponding degree of degeneracy. "
>
> $\textbf{Point 2:}$
>  Authors should add DNN training details (optimizer used, learning rate, number of epochs, etc) in the paper.
>
> $\textbf{Reply}$
>
> We add in the section 3.2 the following details of experiments.
>
> "Throughout this work, we use two-layer fully-connected neural network with size $d$-$m$-$d_{out}$. The input dimension $d$ is determined by the training data. The output dimension $d_{out}$ is different for different experiments. The number of hidden neurons $m$ is specified in each experiment. All parameters are initialized by a Gaussian distribution with mean zero and variance specified in each experiment. We use MSE loss trained by full batch gradient descent for 1D fitting problems (Figs. 1, 3(a) and 4), and default Adam optimizer with full batch for others. The learning rate is fixed throughout the training. More details of experiments are shown in Appendix B.
> "
>
> We add in the appendix the following details of experiments.
>
> "For the 1D fitting experiments (Figs. 1, 3(a) and 4), we use tanh as the activation function, mean squared error (MSE) as the loss function. We use the full-batch gradient descent with learning rate 0.005 to train NNs for 300000 epochs. The initial distribution of all parameters follows a normal distribution with a mean of 0 and a variance of $\frac{1}{m^3}$.
>
> For the iris classification experiment (Fig. 3(b)), we use sigmoid as the activation function, MSE as the loss function. We use the default Adam optimizer of full batch with learning rate 0.02 to train for 500000 epochs. The initial distribution of all parameters follows a normal distribution with mean $0$ and variance $\frac{1}{m^6}$.
>
> For the experiment of MNIST classification (Fig. 5), we use ReLU as the activation function, MSE as the loss function. We also use the default Adam optimizer of full batch with learning rate 0.00003 to train for 100000 epochs. The initial distribution of all parameters follows a normal distribution with mean $0$ and variance $\frac{1}{m^6}$."
>
> $\textbf{Point 3:}$
> The paper is nicely written with a comprehensive related work review, though difficult to follow at certain sections. I would suggest authors to revise the writing for better readability and ease of following.
>
> $\textbf{Reply}$
>
> We mainly make the following revisions to improve the readability and clarity of our work.
> (i) Add detailed experimental settings.
> (ii) Clarify our assumptions as well as definition of some important concepts.
> (iii) Add necessary discussions.
>
> For example, we add the following assumption in Section 3.1.
>
> "Assumptions:
>
> (i) $L$-layer ($L\geq 2$) fully-connected NN.
>
> (ii) Training data $S=\{(x_i,y_i)\}_{i=1}^n$, $n\in\mathbb{Z}^+\cup\{+\infty\}$.
>
> (iii) $R_S(\theta)=E_S\ell(f_{\theta}(x),y)$ without explicit regularization term.
>
> (iv) Loss and activation functions are differentiable. Note that, even for functions like ReLU or hinge loss, as long as we uniquely assign a subgradient to their non-differentiable points, all our results still hold."
>
> For clarity, we change all "hyperplane" to "affine subspace" in our work and add the following definition in Section 3.1.
>
> "Definition: (critical submanifold/affine subspace).
>
> A critical submanifold or affine subspace $\mathcal{M}$ is a connected subsubmanifold or affine subspace of the parameter space $\mathbb{R}^M$, such that each $\theta\in\mathcal{M}$ is a critical point of loss with the same loss value."
>
> We add the following discussion about CNN in Section 6.
>
> "Considering convolutional neural networks for example, the quantity that corresponds to width of fully-connected NNs is channel. Similar to one-step or multi-step embedding, we can introduce a feature splitting operation, i.e., increase the number of channels by splitting all neurons sharing one convolution kernel with the same $\alpha$, which can be proven to preserve the output function, representation as well as the criticality. Thereby, embedding principle holds in a sense that loss landscape of any CNN contains all critical points of all narrower CNNs whose number of channels in each layer is no more than that of the target CNN. "
>
> $\textbf{Point 4:}$
> Also, some of the figure captions can be improved for better reader comprehension, for example the caption of Figure 4.
>
> $\textbf{Reply}$
>
> We have proofread figure captions and made revisions. For Fig. 4, We add the following details of experiments.
>
> "Empirical diagram of loss landscape of a width-$3$ two-layer tanh NN, i.e., all critical points that width-3 or narrower NNs may get close to during the training under proper initialization. Each black dot at terminal represents a specific set of critical points of loss embedded from critical points of NNs of different widths (blue). These critical points have different loss values (ordinate), degrees of degeneracy (green) and output functions (red solid curves) as labelled in the figure. The blue dots represent the training data. We use the same equal splitting as Fig. 3 to embed critical points of width-1 or width-2 NN to critical points of the width-3 NN and compute the hessian to obtain the corresponding degree of degeneracy. Note that the degree of degeneracy of these critical points computed numerically in this problem coincides with their minimal degree of degeneracy $m-m_0$ in Theorem 2."

---

### Author Response · Authors · 2021-08-09
**Response to all Reviewers**

Dear reviewers,

We thank the reviewers for your thoughtful and insightful comments.  We have addressed every comment, and believe that, taken together, the reviewers' comments have improved the manuscript significantly. To address reviewers' common concerns, we have refined the definition, added experimental setups, and improved the writings and figure captions. We hope that the revised manuscript now satisfies the reviewers' requirements, and hereby submit the revised manuscript for publication.

Sincerely yours,

Authors.

---

### Public Comment · ~Alberto_Bernacchia1 · 2021-11-26
**Very interesting results but theorems have been proved 20 years ago**

I really enjoyed reading this paper, it's very clear and raises some interesting points that are crucial for understanding the performance of deep neural networks.
However, when I saw this paper I realized that very similar results were published at ICML 2021:
https://arxiv.org/abs/2105.12221
Concerning theoretical results, the ICML paper pushes the theory much further and, by tracing back the references therein, it turns out that the theorems published here by Zhang et al have been known at least for 20 years:
https://www.sciencedirect.com/science/article/abs/pii/S0893608000000095
On the positive side, I think the paper by Zhang et al will popularize this topic that I think it's very interesting.

---

> ### Public Comment · ~Zhiqin_Xu1 · 2021-11-28
> **Clarify potentially misleading points in the comment**
>
> Dear Alberto,
>
> 	Thanks for your interest and the comment. We have to clarify some potentially misleading points in your comment and hope you can revise your comment title.
> 	First, the ICML 2021 paper [1] (https://arxiv.org/abs/2105.12221) is revealed on arXiv only 5 days before our paper (https://arxiv.org/abs/2105.14573) [2]. The authors of [1] have contacted us and we have agreed that both papers have obtained very similar results INDENPENDENTLY and both sides would acknowledge each other’s work in the future since it has passed the camera-ready submission deadline.
> 	Second, the work in Fukumizu and Amari (2000) [3] only focuses on networks with only one hidden layer and one output unit. In addition, only one-step case from width-(H-1) to width-H is considered. However, in our paper, we prove the embedding principle for networks with arbitrary layers and arbitrary output units. Therefore, it is UNFAIR to say that our result is known for 20 years. Thank you for showing the reference, we will acknowledge this work in our future studies.
> 	Next, we would like to emphasize that the starting point of our work is different from other works. Our starting point originates from our previous paper “Phase Diagram for Two-layer ReLU Neural Networks at Infinite-width Limit” just published in JMLR (20-1123.pdf (jmlr.org)) , where we identify a highly nonlinear condensed regime far beyond the NTK regime that weights condense in isolated directions during the training. Moreover, neural networks of different width often exhibit similar condensed behavior, e.g., stagnating at similar loss with almost the same output function. The current paper is our attempt to uncover the theoretical structure underlying these experimental phenomena.
> 	Therefore, though the parallel work Simsek et al., (2021) [2] and our work propose similar mappings between NN of different widths, we specifically emphasize the importance of having “simple” critical points in wide NNs by stating and proving EXPLICITLY the Embedding Principle, which provides a clear message and solid theoretical foundation for studying highly nonlinear implicit regularization effect of wide NNs towards “simple” functions. On the other side, Simsek et al., (2021) [2] studies the loss landscape from the perspective of the permutation symmetries. Fukumizu and Amari (2000) [3] focus on elucidating the geometric or topological structure of the parameter space. The critical embedding relation in these other works is treated as a sideline rather than the key point as in our work.
>
>
> [1] Simsek et al., Geometry of the Loss Landscape in Overparameterized Neural Networks: Symmetries and Invariances, ICML, https://arxiv.org/abs/2105.12221, 2021.
> [2] Zhang et al., Embedding Principle of Loss Landscape of Deep Neural Networks (This paper), NeurIPS, https://arxiv.org/abs/2105.14573, 2021.
> [3] Fukumizu and Amari, Local minima and plateaus in hierarchical structures of multilayer perceptrons, 2000.
>
> Yaoyu and Zhiqin

---

> > ### Public Comment · ~Alberto_Bernacchia1 · 2021-12-03
> > **Response**
> >
> > I'm sorry, it seems that my words have been interpreted as an attack of your paper, but that was not my goal. Honestly I don't care who takes credit for whatever theorem, I just thought that readers need to know about relevant work that has not been cited. I really think that you should have cited that work in the final version of your paper since, from your response, it sounds like you were aware of it at the time of camera ready submission (at least the ICML 2021 paper, but the paper from 1999 is cited there so...)
> > Besides that, I think your paper is very interesting and it's one of the best I have seen at NeurIPS 2021 so far.

---

### Decision · Program_Chairs · 2021-09-27

**Decision:**

Accept (Spotlight)

**Comment:**

This paper studies critical points in the loss landscape of deep neural networks and proves an "embedding principle", which the reviewers find novel and interesting. This could provide a framework towards deeper understanding of deep learning loss landscape. Most of the questions were raised regarding to the presentation and clarification. The reviewers are overall satisfied with the authors' responses and unanimously recommended acceptance.